# Learning Structured Representations by Embedding Class Hierarchy with Fast Optimal Transport

**Siqi Zeng**[1]   **Sixian Du**[2]   **Makoto Yamada**[3]   **Han Zhao**[1]

[1]University of Illinois, Urbana-Champaign    [2]Stanford University

[3]Okinawa Institute of Science and Technology

`{siqi6, hanzhao}@illinois.edu; dusixian@stanford.edu,`
`makoto.yamada@oist.jp`

## Abstract

To embed structured knowledge within labels into feature representations, prior work (Zeng et al., 2022) proposed to use the **Co**phenetic **C**orrelation **C**oefficient (CPCC) as a regularizer during supervised learning. This regularizer calculates pairwise Euclidean distances of class means and aligns them with the corresponding shortest path distances derived from the label hierarchy tree. However, class means may not be good representatives of the class conditional distributions, especially when they are multi-mode in nature. To address this limitation, under the CPCC framework, we propose to use the Earth Mover's Distance (EMD) to measure the pairwise distances among classes in the feature space. We show that our exact EMD method generalizes previous work, and recovers the existing algorithm when class-conditional distributions are Gaussian. To further improve the computational efficiency of our method, we introduce the **O**ptimal **T**ransport-CPCC family by exploring four EMD approximation variants. Our most efficient OT-CPCC variant, the proposed Fast FlowTree algorithm, runs in linear time in the size of the dataset, while maintaining competitive performance across datasets and tasks. The code is available at `https://github.com/uiuctml/OTCPCC`.

## 1 Introduction

Supervised classification problems have been a cornerstone in machine learning. While traditional classification methods treat class labels as independent entities, there has been a growing interest in leveraging semantic similarity among classes to improve model performance while following existing label dependency. This is particularly important in fine-grained classification tasks where labels are inherently living on a hierarchy, such as in the CIFAR (Krizhevsky & Hinton, 2009) and ImageNet (Deng et al., 2009) datasets. To capture the hierarchical relationship between classes, Zeng et al. (2022) made a significant contribution by embedding the tree metric of a given class hierarchy into the feature space. They introduced a regularizer, the Cophenetic Correlation Coefficient (CPCC), to correlate the tree metric with the Euclidean distance in the class-conditioned representations. This approach of encoding hierarchical relationship in the latent space not only led to more interpretable features compared to other hierarchical classification methods (Shen et al., 2023), but also improved some generalization performance across multiple datasets.

Despite the value of this framework for embedding class hierarchies, it presents a notable limitation. Specifically, Zeng et al. (2022) uses the $\ell_2$ norm between two class Euclidean centroids, or equivalently, maximum mean discrepancy with the linear kernel, as the distance of two class distributions. However, this approach overlooks the difference of samples within the same class, and thus cannot fully capture the global structural information given pairs of class-conditional distributions, especially when they are multi-modal. On the other hand, the use of Optimal Transport (OT) as an alternative to Euclidean centroids has first gained attention in the context of clustering, where Euclidean centroids neglect the intrinsic geometry of data distributions (Cuturi & Doucet, 2014). For the same motivation, we propose addressing the limitation of $\ell_2$-CPCC (Zeng et al., 2022), by integrating the OT distance,

also known as the Wasserstein distance, and its variants, as a more effective metric for measuring the distance between class distributions in the problem of embedding hierarchical representations.

Our contributions are as follows: (i) We first generalize the $\ell_2$-CPCC with EMD, and propose a family of OT-CPCC algorithm to address the limitation of using $\ell_2$ centroids that misrepresents class relationships. We provide a detailed comparative analysis of $\ell_2$-CPCC versus OT-CPCC about their forward and backward computation. (ii) We propose a novel linear time OT approximation algorithm, FastFT, and apply it in the CPCC framework to learn hierarchical representations that is significantly more efficient than the cubic time exact EMD solution. (iii) We empirically analyze the advantage of OT-CPCC over $\ell_2$-CPCC across a wide range real-world datasets and tasks.

## 2 PRELIMINARIES

**Notation and Problem Setup** Throughout the paper we focus on the supervised $k$-class classification problem. We denote a sample of input $x$ with label $y$ living in the space of $\mathcal{X} \subseteq \mathbb{R}^p$ and $\mathcal{Y} = [k] := \{1 \ldots, k\}$. The dataset $\mathcal{D}$ consists of $N$ data pairs $\{(x_i, y_i)\}_{i=1}^N$.

Given the label space $\mathcal{Y}$, we have a corresponding *label tree* $\mathcal{T}$ that characterizes the relationship between different subsets of $\mathcal{Y}$. In particular, each node of $\mathcal{T}$ is associated with a subset of $\mathcal{Y}$: each leaf node $v$ is associated with a particular class in $\mathcal{Y}$; any internal node corresponds to the subset that is the union of its leaf children classes; the root node of $\mathcal{T}$ corresponds to the full $\mathcal{Y}$. The metric function of $\mathcal{Y}$ is called *tree metric* $t : \mathcal{Y} \times \mathcal{Y} \to \mathbb{R}_+$ which measures the similarity of two class labels $u, v$ based on the shortest path distance of two *leaf nodes* on $\mathcal{T}$. Intuitively, the tree $\mathcal{T}$ could be understood as a dendrogram of $\mathcal{Y}$ that characterizes the similarity of different classes.

Let $\mathcal{Z} \subseteq \mathbb{R}^d$ be the representation/feature space. A deep network $h$ is the composition of two functions $h_{w,\theta} = g_w \circ f_\theta : \mathcal{X} \to \Delta_k$ where $\Delta_k$ is a $k - 1$ dimensional probability simplex. The feature encoder $f_\theta$ contains all layers until the penultimate layer of a neural network. On top of that, $g_w : \mathcal{Z} \to \Delta_k$ is a linear classifier with weights $w$. Let $q$ be the one-hot encoding vector of $y$. The parameters $\theta$ and $w$ are learned by minimizing $\mathcal{L}$ combining the cross entropy loss $\ell_{CE}$ with an optional regularization term $\mathcal{R}$:

$$\mathcal{L}(x, y; w, \theta) = \sum_{i=1}^k -q_i \log h_{w,\theta}(x)_i + \lambda \mathcal{R}(x, y, \theta). \tag{1}$$

The hyperparameter $\lambda$ is the regularization factor which controls the strength of $\mathcal{R}$. The goal of the problem is to embed the class hierarchical relationship in the feature space. At a high level, the distance of class-conditioned representations should approximate the tree metric of classes in $\mathcal{T}$. This extra constraint can be quantitatively characterized by CPCC as the regularizer term $\mathcal{R}$.

### 2.1 COPHENETIC CORRELATION COEFFICIENT

The Cophenetic Correlation Coefficient (CPCC) (Sokal & Rohlf, 1962) was introduced as a regularization term in Zeng et al. (2022) to embed the class hierarchy $\mathcal{T}$ in the feature space. CPCC has the same formulation as Pearson's correlation coefficient that measures the linear dependence between two metrics $t$ and $\rho$:

$$\text{CPCC}(t, \rho) = \frac{\sum\limits_{u < v} (t(u, v) - \bar{t})(\rho(u, v) - \bar{\rho})}{(\sum\limits_{u < v} (t(u, v) - \bar{t})^2)^{-1/2} (\sum\limits_{u < v} (\rho(u, v) - \bar{\rho})^2)^{-1/2}}, \tag{2}$$

where $\rho$ is a distance function between $u, v$ on feature space $\mathcal{Z}$, and $\bar{t}, \bar{\rho}$ are the average of each metric. Let $\mathcal{D}_u = \{(x_i^u, u)\}_{i=1}^m, \mathcal{D}_v = \{(x_i^v, v)\}_{i=1}^n \subseteq \mathcal{D}$ be datasets with the same class label $u, v$. When $\ell_2$-CPCC is used as a regularizer in Eq. 1,

$$\rho(u, v) := \left\| \frac{1}{m} \sum_{(x_i^u, u) \in \mathcal{D}_u} f(x_i^u) - \frac{1}{n} \sum_{(x_j^v, v) \in \mathcal{D}_v} f(x_j^v) \right\|_2,$$

which is the $\ell_2$ distance of the mean of the two class-conditional features. We call $z_i^u$ as the $i$-th feature vector with label $u$, when the context is clear, we drop the superscript $u$ for convenience. By

| **Algorithm 1** Greedy Flow Matching | **Algorithm 2** Bottom-up Tree Flow Matching |
|---|---|
| **Input:** number of features $m, n$, weights $a = (a_1, \ldots, a_m)$, $b = (b_1, \ldots, b_n)$ | **Input:** $m, n, a, b$, flow tree $\mathcal{T}$ |
| 1: $\boldsymbol{P} = \text{Array}[m \times n]$ | 1: $\boldsymbol{P} = \text{Array}[m \times n]$ |
| 2: $i = 0$ | 2: **for** height $h$ from 0 to $\text{height}(\mathcal{T})$ **do** |
| 3: $j = 0$ | 3:    **for** each node $v$ at height $h$ **do** |
| 4: **while** $i < m$ **and** $j < n$ **do** | 4:      **if** $v$ is a leaf **then** |
| 5:    $\boldsymbol{P}[i,j] \leftarrow \min(a[i], b[j])$ | 5:       Pass its flow to its parent |
| 6:    $a[i] \mathrel{-}= \boldsymbol{P}[i,j]$ | 6:      **else** |
| 7:    $b[j] \mathrel{-}= \boldsymbol{P}[i,j]$ | 7:       $C_a(v) = \{z : a(z) > 0\}, C_b(v) = \{z : b(z) > 0\}$, $z$ is the leaf children of $v$ |
| 8:    **if** $a[i] == 0$ **then** $\quad i \mathrel{+}= 1$ | 8:       **while** $\exists z_a \in C_a(v)$ **and** $z_b \in C_b(v)$ **do** |
| 9:    **if** $b[j] == 0$ **then** $\quad j \mathrel{+}= 1$ | 9:        $\eta = \min(a(z_a), b(z_b))$ |
| | 10:        $\boldsymbol{P}(z_a, z_b) \mathrel{+}= \eta, a(z_a) \mathrel{-}= \eta, b(z_b) \mathrel{-}= \eta$ |
| | 11:      Pass non-zero unmatched flows in either $a$ or $b$ to its parent |
| **Output:** $\boldsymbol{P}$ | **Output:** $\boldsymbol{P}$ |

maximizing CPCC ($\lambda < 0$), $\rho$ follows the relation in $t$: classes sharing a close common ancestor will be grouped together while classes with a higher common ancestor are pulled away with respect to Euclidean distance.

## 2.2 Exact Earth Mover's Distance

**Earth Mover's Distance** (EMD), or Optimal Transport distances, aims to find the optimal transportation plan between two discrete measures. For $\mathcal{D}_u$, let $a = (a_1, \ldots, a_m) \in \Delta_m$ which is not necessarily uniform. Let $b$ defined similarly for $\mathcal{D}_v$, and $\delta_z$ is the Dirac delta at $z$. Define two discrete measures as $\mu = \sum_{i=1}^{m} a_i \delta_{z_i}, \nu = \sum_{j=1}^{n} b_j \delta_{z_j}$. Then, EMD between probability distributions defined over $\mathcal{Z}$, is the objective value of the following linear program:

$$\min_{\boldsymbol{P}} \langle \boldsymbol{P}, \boldsymbol{D} \rangle, \quad \text{s.t. } \boldsymbol{P}\mathbf{1}_n = a, \ \boldsymbol{P}^\top \mathbf{1}_m = b. \tag{3}$$

In Eq.3, $\langle \cdot, \cdot \rangle$ is the Frobenius inner product, $\mathbf{1}_N$ is the $N$-dimensional all ones vector, $\boldsymbol{D}$ is the $m \times n$ pairwise distance matrix where $\boldsymbol{D}_{ij} = \|z_i - z_j\|$, and $\boldsymbol{P}$ is called the flow matrix in the computer science literature, or optimal transport plan in the OT literature, the same size with $\boldsymbol{D}$ with nonnegative entries. Intuitively, this is a minimum cost flow problem. Both source and target flow sum to 1, and we want to find the optimal transportation plan $\boldsymbol{P}$ with the minimum cost with respect to ground similarity $\boldsymbol{D}$. The norm $\|\cdot\|$ depends on the ground metric of $\boldsymbol{D}$. We use the $\|\cdot\|_2$ norm as the ground metric throughout this paper.

When the dimension of the feature is 1, fast computation of the EMD distance exists (Houry et al., 2024). Instead of solving a linear program in Eq.3, when $m = n$, $a_i = b_i = 1/n$ for all $i$, the closed form solution for **1d EMD** is: $\text{EMD}_{1d}(\mu, \nu) = \frac{1}{n} \sum_{i=1}^{n} \|z_{(i)} - z'_{(i)}\|$, and $z_{(i)}$ is the $i$-th order statistic of the sequence $\{z_1, \cdots z_n\}$ after sorting. When $m \neq n$, $\text{EMD}_{1d}$ generalizes to solving inverse CDF of $\mu, \nu$ (Peyré & Cuturi, 2019) which requires approximation. To avoid this approximation, *after sorting*, we can compute the flow matrix $\boldsymbol{P}$ (Hoffman, 1963) efficiently via a greedy algorithm in linear time (Alg.1) and use the inner product objective of EMD in Eq.3 to compute the result. This greedy flow matching algorithm that generates the optimal transport plan can be used for any $m, n$, including the special case when $m = n$, and fit any non-uniform weights of $a, b$.

## 3 CPCC with Optimal Transport

### 3.1 OT-CPCC Family

Despite its simplicity, *the use of class centroids $\ell_2$ distance as a measure of class distances has two limitations.* First, since there is no guarantee of the distribution of class conditional features, the class mean difference may not accurately reflect the average pairwise Euclidean distance between all the pairs of sample. For example, in Fig.1, class $v$ is a non-unimodal (bi-modal in this example) distribution, which is very likely to happen when class $v$ is an abstract concept, such as coarse classes like household furniture in CIFAR100 (Krizhevsky & Hinton, 2009) or the collection of diverse dog breeds in ImageNet Deng et al. (2009). The blue $\ell_2$ line can be a biased estimation of the average of

red lines. We will see later (c.f. Sec.3.3) that the direction of blue vector plays an important role in the feature geometry in $\ell_2$-CPCC, and therefore we misrepresent the class distribution in $\ell_2$-CPCC. Second, it is known (Linial et al., 1994) that, for certain trees such as a basic star graph with three edges, it is impossible to embed tree metric $t$ into $\ell_2$ exactly, no matter how large feature dimension $d$ is. It implies if we use the Euclidean *centroids* to represent a class, we can never reach the optimal CPCC value 1 during training, which always cause some loss of hierarchical information in the learning of structured representation.

How to overcome the above limitations? Instead of finding a one-to-one correspondence of a label and a point estimate of a class, we represent a label class with all *samples* that belong to the class, which leads to EMD-CPCC. We call the set of data-weight pairs containing $m$ class-$u$ conditioned features as $\mu = \{(z_i, a_i)\}_{i=1}^m$ and define $\nu$ similarly for the dataset containing $n$ class-$v$ conditioned features. The weight $a_i$ can be set as uniform $(1/m)$ by default, or non-uniform if external prior knowledge is available. We propose EMD-CPCC by implementing $\rho(u, v) := \text{EMD}(\mu, \nu)$. In Eq.3, $\boldsymbol{P}$ depends on $\boldsymbol{D}$ derived from each *sample* in two datasets. Besides, EMD is actually a generalization of $\ell_2$ method:

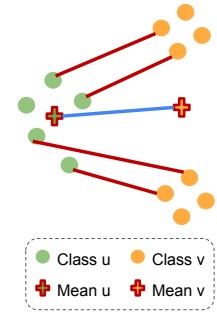

**Proposition 3.1** (EMD reduces to $\ell_2$ between means of Gaussian with same covariance). $\text{EMD}(\mathcal{N}(\mu_z, \Sigma), \mathcal{N}(\mu_{z'}, \Sigma)) = \|\mu_z - \mu_{z'}\|$.

Figure 1: Comparison between EMD (weighted sum of red lines) and the class mean $\ell_2$ distance (blue line).

Although EMD has many appealing properties, the computational cost of EMD is high, where the time complexity of linear programming is cubic time with respect to the data size. Many following work provides approximation methods to reduce the computational cost, such as Sinkhorn (Cuturi, 2013), Sliced Wasserstein Distance (SWD) (Bonneel et al., 2015), Tree Wasserstein Distance (TWD) (Yamada et al., 2022; Le et al., 2019; Takezawa et al., 2021), and FlowTree (FT) (Backurs et al., 2020) (detailed discussion of approximation methods in App.A). Similarly, we can replace $\rho$ with all other EMD approximation methods, and we call all of them as the **OT-CPCC family**. As a side remark, since all approximation methods provide a $\boldsymbol{P}$ satisfying constraints in Eq.3 and since EMD is lower bounded by $\ell_2$ (see proof of Prop.3.1 in App.C), all OT-CPCC methods are still lower bounded by $\ell_2$.

## 3.2 Fast FlowTree

**FlowTree** (Backurs et al., 2020) is a tree-based method where the structure of the tree $\mathcal{T}$ is learned using QuadTree (Samet, 1984) to approximate the Euclidean space. Once the tree has been built, it firsts compute the optimal flow matrix $\boldsymbol{P}_t$ using $t$ as the ground metric with a linear time bottom-up as shown in Alg.2 (Kalantari & Kalantari, 1995) (see App.Fig.8 in for visualization). Then, it estimates EMD with $\langle \boldsymbol{P}_t, \boldsymbol{D}_{\ell_2} \rangle$. Given the tree, the time complexity of computing the FT is $O((m + n)d \log(d\Phi))$ where $\Phi$ is the maximum Euclidean distance of any two feature vectors.

**Our Fast FlowTree** The main computational bottleneck of FT is the construction of the tree. However, in our case, we have access to $\mathcal{T}$ given as prior knowledge, and thus we can skip the tree construction step. To apply tree-based EMD approximation methods, instead of building a QuadTree in FT, we first construct an *augmented tree* by extending the leaves of the original label tree $\mathcal{T}$ with samples. Specifically, since each leaf node in $\mathcal{T}$ corresponds to a class, we extend each leaf node by adding all samples from the corresponding class, as demonstrated in Fig.2. We set all extended edges to have edge weight 1, and assign the instance weight $a_i$ for each data point. The resulting algorithm that applies Alg.2 to the *augmented* label tree is called **Fast FlowTree** (FastFT).

**Key Insight** that allows FastFT to be more efficient than FT is that the tree metric between any two data samples from different classes is always identical in the augmented tree $\mathcal{T}$, which allows us to reduce the computation of $\boldsymbol{P}_t$ to 1d OT greedy flow matching using Alg.1. Now we can prove that Alg.1 and Alg.2 are equivalent given our constructed augmented tree in Fig.2, and the time complexity of FastFT is $O((m + n)d)$, which is optimal among all OT variants.

**Theorem 3.2** (Correctness of Fast FlowTree). *For any augmented label tree $\mathcal{T}$, Alg. 1 and Alg. 2 return the same EMD approximation. The Fast FlowTree can be computed in $O((m + n)d)$.*

| Method | Time Complexity | $\frac{\partial \mathcal{L}}{\partial \theta}$ exact | Use Tree |
|---|---|---|---|
| $\ell_2$ | $O((nk+k^2)d)$ | ✓ | N/A |
| EMD | $\tilde{O}(n(d+n^2\log n))$ | ✓ | |
| Sinkhorn | $\tilde{O}(n^2(d+I))$ | ✓ | |
| TWD | $\tilde{O}(n^2dI)$ | | ✓ |
| SWD | $\tilde{O}(np(d+\log n))$ | | |
| FT | $\tilde{O}(nd\log(d\Phi))$ | | ✓ |
| FastFT | $\tilde{\Theta}(nd)$ | | ✓ |

Table 1: Time complexity comparison of different CPCC methods. Let $I$ be the number of iterations for iterative methods, $p$ be the number of projections for SWD, $\Phi$ be the maximum Euclidean distance of any two feature vectors. We use $\tilde{\ }$ to represent a factor of $k^2$. For simplicity we assume $m = n$. In the batch learning setting, $d$ can be much larger than $n$ particularly for the fine-grained classification problems. See App.I for detailed analysis.

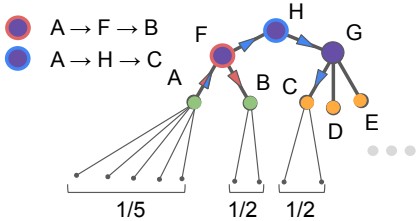

Figure 2: An example of augmented $\mathcal{T}$. Leaves become samples of each class, and each sample is assigned some weight (ex., uniform) within a class. Whenever we call FastFT-CPCC, only a subtree rooted at the lowest common ancestor of a pair of class label will be used. For example, we use subtree rooted at $F$ to compute optimal flow of samples with label $A, B$, and subtree rooted at $H$ for samples with label $A, C$.

Please refer to App.D for proof details of Theorem 3.2. Note that all OT-CPCC methods inherently need $\Omega((m+n)d)$ because we access each of $m+n$ data point once and look at each entry in $d$-dimensions. Additionally, FastFT can be applied to any label tree with no restriction on structure or edge weights. To control the importance of data points, the flexibility of FastFT can be achieved by tuning $a_i$.

**Comparison of OT-methods on Synthetic Dataset**   To further understand the behavior of different OT approximation methods, we present experiments on synthetic datasets in Fig.3 (see App.J for more details). We first measured the computation time for different OT methods across datasets of varying sizes. Second, we evaluated the error of different OT approximation methods, along with the $\ell_2$ distance of the mean of two datasets, in approximating EMD. Two different scenarios were considered, either both distributions were Gaussian or non Gaussian. Under Gaussian distribution, the OT methods' results were closely aligned with the $\ell_2$ method, except for SWD, which supports Prop.3.1 and implies the similarity in performance between OT and $\ell_2$-CPCC under Gaussian distributions. In non-Gaussian distribution scenario, notable differences emerged between OT methods and $\ell_2$, where SWD is closer to $\ell_2$ measurement. FastFT demonstrated exceptional advantage as an approximation method, outperforming Sinkhorn in both computational speed and precision.

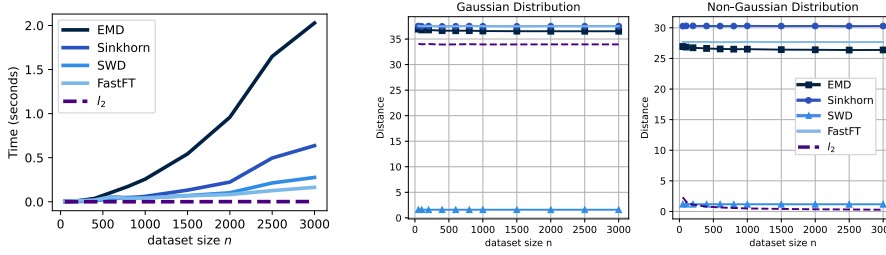

Figure 3: Efficiency and approximation error comparison of OT methods on synthetic datasets.

## 3.3 DIFFERENTIABILITY ANALYSIS OF OT-CPCC

Since we use some non-differentiable algorithms during training (ex., linear programming in EMD, sorting in SWD, tree construction in TWD, greedy algorithm in FT and FastFT), it is unclear how does the backpropagation algorithm work exactly in this problem. Thanks to the Danksin's theorem (Danskin, 1966) and the structure of the underlying optimization problem, we can still

efficiently compute the gradients of OT-CPCC. For simplicity, let us consider the case of EMD-CPCC, i.e., $\rho = $ EMD exactly. The other approximate variants can be analyzed in a similar way.

The underlying challenge of the problem is to take derivatives of the optimal value of a minimization problem. We state the differentiability section of Danskin's theorem formally and prove a lemma:

**Theorem 3.3** (Danskin (1966)). *Let $V(x) = \max_{z \in \mathcal{C}} f(x, z)$, and $Z_0(x)$ is the set of $z$ that achieve the maximum of $V$ at $x$. If $f : \mathbb{R}^n \times \mathcal{C} \to \mathbb{R}$ is continuous and $\mathcal{C}$ is compact, then the subgradient of $V$ is the convex hull of set $\{\frac{\partial f(x,z)}{\partial x} : z \in Z_0\}$. Particularly, if $Z_0 = \{z^*\}$, $\frac{\partial V}{\partial x} = \frac{\partial f(x,z^*)}{\partial x}$.*

**Lemma 3.4.** *Denote $\boldsymbol{P}^*$ as the optimal flow of EMD problem, then $\frac{\partial \rho_{EMD}}{\partial \theta} = \boldsymbol{P}^* \cdot \frac{\partial \boldsymbol{D}}{\partial \theta}$.*

*Proof.* We rewrite the objective function as a maximization problem: $\rho_{\text{EMD}} = -\max_{\boldsymbol{P} \in \mathcal{C}} -\langle \boldsymbol{P}, \boldsymbol{D} \rangle$. Roughly speaking, Danksin's theorem states that the derivative of maximization problems can be evaluated at the optimum. We can get (sub-)differential of $\rho_{\text{EMD}}$ with respect to $\theta$ with one step of chain rule:

$$\frac{\partial \rho_{\text{EMD}}}{\partial \theta} = \frac{\partial \rho_{\text{EMD}}}{\partial \boldsymbol{D}} \cdot \frac{\partial \boldsymbol{D}}{\partial \theta} = -\frac{\partial - \langle \boldsymbol{P}^*, \boldsymbol{D} \rangle}{\partial \boldsymbol{D}} \cdot \frac{\partial \boldsymbol{D}}{\partial \theta} = \boldsymbol{P}^* \cdot \frac{\partial \boldsymbol{D}}{\partial \theta},$$

where the last term comes from the original flow of backpropagation containing the gradients with respect to network parameters. The key takeaway is the gradient of $\rho_{\text{EMD}}$ w.r.t. the network parameters does not require the gradient from $\boldsymbol{P}$ in the EMD setting, although $\boldsymbol{P}$ depends on $\theta$. ∎

**Differentiability for Other OT-CPCC Methods**    The application of Danskin's relies on the optimality of $\boldsymbol{P}^*$. How about the other approximation methods? The differentiability is not a problem in Sinkhorn as the iterative algorithm only involves matrix multiplication. But for the SWD, TWD, FT, the approximate solutions (the flow matrix $\boldsymbol{P}$) for these methods are not explicitly formulated as solutions to an optimization problem. Following EMD, except for the Sinkhorn variant, our method is to treat flow matrix $\boldsymbol{P}$ always as a constant for these approximation methods, and stop the gradient through $\boldsymbol{P}$. For example, in Fast FlowTree, since the tree flow matrix $\boldsymbol{P}$ depends only on the structure of the tree based on external information and not on the model parameters $\theta$, we have $\frac{\partial \rho_{\text{FastFT}}}{\partial \theta} := \boldsymbol{P}_{\text{FastFT}} \cdot \frac{\partial \boldsymbol{D}}{\partial \theta}$.

To provide a more fine-grained understanding of the proposed method, in what follows we analyze the gradients of the proposed CPCC variants and compare them with the $\ell_2$-CPCC.

**Gradient of $\ell_2$-CPCC**    If we want to know the difference between $\ell_2$-CPCC and EMD-CPCC, assume the same input and the same network parameters, ignoring the cross-entropy loss, the only difference of the gradient starts from the distance calculation between two classes. Assume representations from two classes are $Z \in \mathbb{R}^{m \times d}, Z' \in \mathbb{R}^{n \times d}$.

$$\frac{\partial \rho_{l_2}}{\partial Z} = \frac{\partial}{\partial Z} \|\mu_z - \mu_{z'}\| = \frac{\mathbf{1}(\frac{1}{m}\mathbf{1}^\top Z - \frac{1}{n}\mathbf{1}^\top Z')}{m \left\| \frac{1}{m} Z^\top \mathbf{1} - \frac{1}{n} Z'^\top \mathbf{1} \right\|}.$$

Let $i$-th row of $Z$ be $Z_i$. Each row of $\frac{\partial \rho_{l_2}}{\partial Z}$ is the same, so

$$\frac{\partial \rho_{l_2}}{\partial Z_{i\cdot}} = \frac{\mu_z - \mu_{z'}}{m \|\mu_z - \mu_{z'}\|}.$$

**Gradient of EMD-CPCC**    Let $\boldsymbol{D} \in \mathbb{R}^{m \times n}$ be the cost matrix and $\boldsymbol{P}$ is any constant flow matrix:

$$\rho_{\text{EMD}} = \langle \boldsymbol{P}^*, \boldsymbol{D}(Z, Z') \rangle = \sum_{i,j} \boldsymbol{P}_{ij}^* \left\| Z_{i\cdot} - Z'_{j\cdot} \right\|.$$

Let's calculate the derivative by rows of $Z$. We only keep the terms related to $Z_i$.

$$\frac{\partial \rho_{\text{EMD}}}{\partial Z_{i\cdot}} = \frac{\partial}{\partial Z_{i\cdot}} \sum_j \boldsymbol{P}_{ij}^* \left\| Z_{i\cdot} - Z'_{j\cdot} \right\| = \sum_{j=1}^{n} \boldsymbol{P}_{ij}^* \frac{Z_{i\cdot} - Z'_{j\cdot}}{\left\| Z_{i\cdot} - Z'_{j\cdot} \right\|}.$$

Note that since $\boldsymbol{P}^*$ is the optimal solution of a linear program, it is a sparse matrix with at most $m + n - 1$ non-zero entries (Peyré et al., 2019, Proposition 3.4). As a comparison to the gradient of $\ell_2$-CPCC, which assigns equal importance to all data points and only depends on the direction of class centroids, the gradient of EMD-CPCC assigns different importance to each pair of data points and depends on the pairwise transport probability between data points. From this analysis, we can see that the gradient provided by EMD-CPCC is more fine-grained and informative than that of $\ell_2$-CPCC.

## 4 EXPERIMENTS

For a fair comparison, we closely aligned our implementation with Zeng et al. (2022) to ensure consistency. This section will cover dataset benchmarks, the multi-modality of features, downstream hierarchical classification retrieval, and interpretability analysis. A detailed description of training hyperparameters is provided in App.F, along with additional experiments, including the effect of label hierarchy structure, label hierarchy tree weights, optimal transport flow settings (distribution of $a$ in Eq.3), and comparisons with non-CPCC hierarchical methods, in App.H. Throughout this section, the Flat objective is the standard cross entropy loss in Eq.1 with $\lambda = 0$ without tree information.

Table 2: Dataset Statistics.     Table 3: Preciseness of hierarchical information.

| Dataset | # train | # test |
|---|---|---|
| *CIFAR10* | 50K | 10K |
| *CIFAR100* | 50K | 10K |
| *INAT* | 500K | 100K |
| *L17* | 88K | 3.4K |
| *N26* | 132K | 5.2K |
| *E13* | 334K | 13K |
| *E30* | 307K | 12K |

| Dataset | Objective | TestCPCC | Dataset | Objective | TestCPCC |
|---|---|---|---|---|---|
| *CIFAR10* | Flat | 58.83 (8.80) | *L17* | Flat | 47.63 (1.18) |
| | $\ell_2$ | 99.94 (0.02) | | $\ell_2$ | 92.30 (0.35) |
| | FastFT | **99.95 (0.00)** | | FastFT | 91.71 (0.15) |
| | EMD | 99.95 (0.01) | | Sinkhorn | 91.77 (0.44) |
| | Sinkhorn | 99.94 (0.01) | | SWD | **95.10 (0.51)** |
| | SWD | 99.98 (0.00) | | | |
| *CIFAR100* | Flat | 22.13 (0.84) | *N26* | Flat | 29.77 (0.53) |
| | $\ell_2$ | 83.08 (0.11) | | $\ell_2$ | 93.03 (0.32) |
| | FastFT | **93.85 (0.05)** | | FastFT | 92.68 (0.21) |
| | EMD | 89.60 (0.34) | | Sinkhorn | **93.31 (0.24)** |
| | Sinkhorn | 93.81 (0.10) | | SWD | 92.40 (0.05) |
| | SWD | 73.17 (2.66) | | | |
| *INAT* | Flat | 31.97 (0.00) | *E13* | Flat | 49.20 (0.86) |
| | $\ell_2$ | 67.01 (0.04) | | $\ell_2$ | **92.02 (0.08)** |
| | FastFT | **73.20 (0.03)** | | FastFT | 91.97 (0.20) |
| | EMD | 65.80 (0.01) | | Sinkhorn | 91.30 (0.57) |
| | Sinkhorn | 73.20 (0.04) | | SWD | 88.32 (0.50) |
| | SWD | 50.88 (0.03) | | | |
| | | | *E30* | Flat | 51.33 (0.09) |
| | | | | $\ell_2$ | **93.37 (0.19)** |
| | | | | FastFT | 91.81 (0.44) |
| | | | | Sinkhorn | 91.89 (0.61) |
| | | | | SWD | 89.88 (0.68) |

Figure 4: Hierarchical data split. Source and target dataset share the same coarse labels but different fine labels.

### 4.1 DATASETS

We conducted experiments across 7 diverse datasets, *CIFAR10, CIFAR100* (Krizhevsky & Hinton, 2009), *BREEDS* benchmarks (Santurkar et al., 2021) including four settings LIVING17 (*L17*), ENTITY13 (*E13*), ENTITY30 (*E30*), NONLIVING26 (*N26*), and iNaturalist-mini (Van Horn et al., 2018) (*INAT*) to thoroughly evaluate the efficacy and influence of OT-CPCC methods. The dataset statistics is shown in Tab.2. Note that *BREEDS* and *INAT* are larger scale datasets with high resolution images. As subsets of ImageNet, *BREEDS* inherit a more complicated label hierarchy and more realistic, possibly multi-modal class distributions. The major difference from ImageNet is that *BREEDS*'s class hierarchy is manually calibrated so that it reflects the visual hierarchy instead of the WordNet semantic hierarchy in the original ImageNet, which removes ill-defined hierarchical relationships unrelated to vision tasks.

### 4.2 EXISTENCE OF MULTI-MODE CLASS-CONDITIONED FEATURE DISTRIBUTION

To verify our hypothesis in Fig.1, we aim to determine whether class-conditioned clusters exhibit a multi-modal structure in practice. We track the class-conditioned features during training: at each epoch, for each class-conditioned feature cluster, we fit a Gaussian Mixture Model (GMM) with the number of components ranging from 1 to 10. We report the optimal number of components based on the lowest Akaike Information Criterion (AIC), averaged across all fine-level class labels. *If the best number of Gaussian components exceeds 1, this may indicate the presence of a multi-modal structure in the feature distributions*, assuming a sufficiently large batch size. From Fig.5, with default training hyperparameters in App.F, we observe that this phenomenon occurs across most datasets for both $\ell_2$-CPCC and OT-CPCC training, suggesting that using optimal transport distances in CPCC computation is more appropriate.

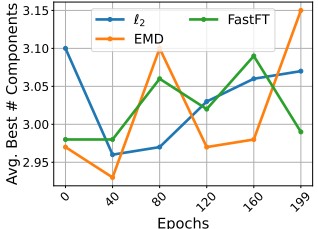

| Dataset | # Comp. |
|---------|---------|
| *CIFAR10* | 9.90 |
| *CIFAR100* | 3.07 |
| *INAT* | 3.48 |
| *BREEDS* | 1.25 |

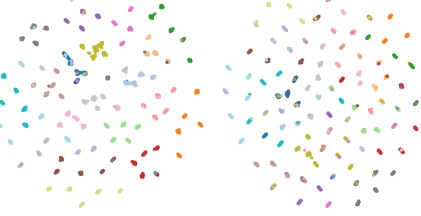

Figure 5: The average best number of components of learned features (left) across training epochs on CIFAR100 for different CPCC methods (right) for the $\ell_2$-CPCC final trained features for all datasets.

Figure 6: CIFAR100 Euclidean UMAP for $\ell_2$-CPCC (left) vs. EMD-CPCC (right). Each cluster represents a fine label and we color data sample by coarse labels.

### 4.3 HIERARCHICAL CLASSIFICATION AND RETRIEVAL

Table 4: Generalization performance on coarse level. The prefix of metric represents the split of test set: **s** = source, **t** = target. The average number of three seeds are reported. See the detailed version including standard deviation in Tab.16,17 and performance on each BREEDS setting is in Tab.18,19. Due to the computation complexity, we do not report EMD results on training BREEDS from scratch.

| Dataset | Objective | sAcc | tAcc | sMAP | tMAP | Dataset | sAcc | tAcc | sMAP | tMAP |
|---------|-----------|------|------|------|------|---------|------|------|------|------|
| *CIFAR10* | Flat | 99.58 | 87.30 | 99.22 | 89.66 | *INAT* | 94.63 | 38.81 | 70.41 | 34.00 |
| | FastFT | 99.61 | **87.79** | **99.91** | 93.04 | | **94.66** | 39.43 | 72.90 | 35.63 |
| | EMD | **99.61** | 87.45 | 99.88 | **93.07** | | 94.64 | **41.01** | 73.87 | 35.58 |
| | Sinkhorn | 99.61 | 87.41 | 99.87 | 92.60 | | 94.30 | 36.94 | 68.75 | 34.92 |
| | SWD | 99.56 | 87.63 | 99.36 | 90.12 | | 94.52 | 39.38 | **75.13** | **38.11** |
| *CIFAR100* | Flat | 86.17 | 43.09 | 82.07 | 42.09 | *BREEDS* | 93.85 | 82.43 | 74.47 | 66.50 |
| | FastFT | **86.96** | **44.29** | 91.71 | 42.20 | | 94.25 | 82.89 | **92.84** | **79.69** |
| | EMD | 86.93 | 44.03 | **91.82** | 42.32 | | - | - | - | - |
| | Sinkhorn | 86.93 | 43.99 | 91.61 | **42.45** | | 94.42 | **83.51** | 91.84 | 79.33 |
| | SWD | 86.42 | 43.35 | 87.24 | 42.39 | | 94.41 | 82.96 | 83.58 | 72.38 |

Table 5: Generalization performance on fine level.

| Dataset | Objective | sAcc | tAcc | sMAP | Dataset | sAcc | tAcc | sMAP |
|---------|-----------|------|------|------|---------|------|------|------|
| *CIFAR10* | $\ell_2$ | 96.96 | 55.71 | 99.22 | *INAT* | 88.62 | 26.78 | 56.10 |
| | FastFT | 96.90 | 55.99 | 99.24 | | 88.49 | **27.10** | 56.21 |
| | EMD | **97.05** | 56.12 | 99.24 | | **88.68** | 26.78 | **56.83** |
| | Sinkhorn | 96.95 | 54.89 | 99.27 | | 88.08 | 26.77 | 51.56 |
| | SWD | 96.96 | **59.21** | **99.45** | | 88.46 | 26.78 | 54.19 |
| *CIFAR100* | $\ell_2$ | 80.98 | 23.76 | 77.52 | *BREEDS* | 82.66 | 45.95 | 62.86 |
| | FastFT | 81.09 | 24.71 | 78.72 | | 82.58 | 45.75 | 63.95 |
| | EMD | **81.32** | 23.15 | 78.88 | | - | - | - |
| | Sinkhorn | 80.99 | 23.53 | 78.51 | | **82.99** | **46.87** | 61.54 |
| | SWD | 80.50 | **26.18** | **86.82** | | 82.90 | 46.33 | **76.24** |

**Hierarchical Dataset Split** For downstream classification and retrieval, we split each dataset into source and target subsets, with both further divided into train and test sets, as illustrated in Fig.4. The source and target datasets share the same level of coarse labels but differ in fine labels, where the fine level refers to the more granular class labels that are exactly one level below the coarse labels. For example, in *CIFAR10*, the source set includes the fine labels (`deer`, `dog`, `frog`, `horse`) and (`ship`, `truck`), while the target set includes (`bird`, `cat`) and (`airplane`, `automobile`). Both sets share the same coarse labels: `animal` and `transportation`. We include details of hierarchical construction and splits for other datasets in App.E.

Given the hierarchical splits, we define two evaluation protocols. First, we pretrain on the source-train set and evaluate on the source-test set, where the train and test distributions are i.i.d. Second, we pretrain on the source-train set and evaluate on the target-test set, introducing a subpopulation shift (Santurkar et al., 2021) between the train and test distributions. For each protocol, we conduct classification and retrieval experiments at two levels of granularity.

**Metrics** For classification experiments, we measure performance using accuracy (Acc). At the fine level, source accuracy is computed directly using any pretrained model, while coarse source accuracy is calculated by summing the fine-level probabilities for predictions that share the same coarse parent label. Since the source and target sets share the same coarse labels, coarse target accuracy can be similarly computed using the model pretrained on the source split. For fine target accuracy, after pretraining on the source set, we train a linear classifier on top of the frozen pretrained encoder for one-shot learning on the target-train set and evaluate it on the target-test set. We include the visualization of all types of classification tasks evaluation in Fig.9 in the Appendix.

For retrieval experiments, we use mean average precision (MAP) as the evaluation metric. Given a model pretrained on the source set, we compute the fine or coarse class prototypes by calculating the Euclidean centroid of the class-conditioned features. Using these class prototypes, we compute the cosine similarity score between the prototypes and test points, and evaluate whether test points are most similar to the class prototype with the same label via average precision. Since the source and target sets have different fine labels, we do not include target MAP in Tab.5.

**Discussion** *OT-CPCC outperforms $\ell_2$-CPCC consistently on fine level tasks*: the oracle for the fine level tasks should be Flat, and Zeng et al. (2022) reported that $\ell_2$-CPCC does not perform well for several fine level tasks due to the constraint on coarse level. Tab.5 shows that OT-CPCC methods shows better fine-level performance than $\ell_2$, while maintaining hierarchical information simultaneously. Besides, *OT-CPCC outperforms the Flat baseline consistently on coarse level tasks* as expected in Tab.4, which implies OT-CPCC methods successfully include the additional hierarchical information by optimizing the regularizer. In Fig.6, we visualize the learned hierarchical embeddings using UMAP (McInnes et al., 2018), with Euclidean distance as the similarity metric. We observe that $\ell_2$-CPCC displays a clearer coarse-level grouping pattern, likely due to the direct optimization of CPCC in Euclidean space. However, EMD-CPCC also retains a significant amount of hierarchical information and demonstrates better separation between fine-level classes, which aligns with the results observed in our classification and retrieval tasks.

## 4.4 INTERPRETABILITY EVALUATION

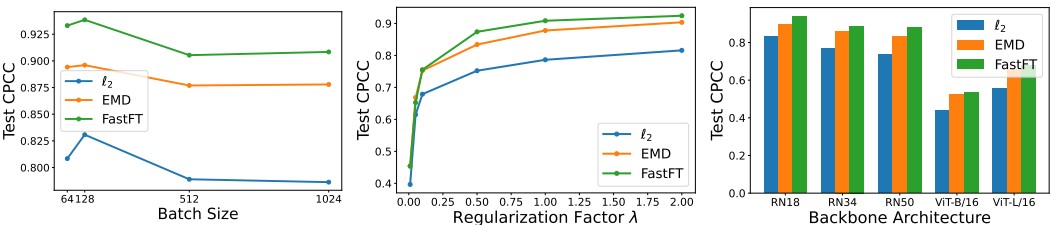

Figure 7: Test CPCC for various batch size, regularization strength, and architecture, from left to right. FastFT consistently performs the best across all settings.

A model is more interpretable when its representations accurately capture hierarchical information. CPCC is a natural measure of the precision of the hierarchical structure embedded in corresponding metric spaces. Unlike rank-based metrics, where only the relative order of distances matters, CPCC not only requires the correct ranking of distances (i.e., fine classes sharing the same coarse class should be closer than fine classes with different coarse parents), but is also influenced by the absolute values of the node-to-node distances, because CPCC is affected by the given label tree weights. Therefore, we report CPCC, a stricter metric, on the test set. In Tab.3, we present $\ell_2$-CPCC for the Flat and $\ell_2$ objectives, and use the corresponding OT distances for the OT-CPCC experiments.

We observe that *OT-CPCC preserves more hierarchical information, leading to better interpretability*. As shown in Tab.3, in 5 out of 7 datasets, an OT-CPCC method achieves the best TestCPCC score. For *E13* and *E30*, OT-CPCC methods slightly underperform compared to $\ell_2$, likely due to the BREEDS representations being less multi-modal than those of other datasets, as indicated in Tab.5.

**Ablation Study**  In our experiments with CIFAR100, shown in Fig.7, we investigated the impact of batch size, the regularization factor $\lambda$ in Eq.1, and architecture on TestCPCC. Given CIFAR100's large number of fine classes, the average sample size per class is relatively small when applying CPCC with a standard batch size of 128. Consequently, there is minimal difference between using the centroid of a few data points, as in $\ell_2$-CPCC, and using pairwise distances for each data point in OT-CPCC. Therefore, larger batch sizes allow OT-CPCC to better capture class distributions, potentially leading to a more significant advantage in the quality of the learned hierarchy. We compared TestCPCC across various settings, including batch sizes of [64, 128, 512, 1024], $\lambda$ values of [0.01, 0.05, 0.1, 0.5, 1, 2], and backbones such as ResNet18, ResNet34, ResNet50 (He et al., 2016), ViT-B/16, and ViT-L/16 (Dosovitskiy, 2020), using $\ell_2$, EMD, or FastFT CPCC. Across all comparisons, we observe that FastFT > EMD > $\ell_2$ in terms of the quality of the learned hierarchy, highlighting the advantage of optimal transport-based methods.

## 5  RELATED WORK

**Learning Hierarchical Representations**  We discuss methods that operate on feature representations, thereby directly influencing the geometry of the representation space when hierarchical information is provided. DeViSE (Frome et al., 2013) is an early multimodal work to align image and text embedding evaluated on a hierarchical precision metric to reflect semantic knowledge in the representation. Ge et al. (2018); Zhang et al. (2016) modify triplet loss to capture class similarity for metric learning or image retrieval. Barz & Denzler (2019); Garg et al. (2022) embed classes on hypersphere based on hierarchical similarity, and similar ideas have been explored in the hyperbolic space (Kim et al., 2023) where tree can be embedded with low distortion. Yang et al. (2022) provides a multi-task proxy loss for deep metric learning. Nolasco & Stowell (2022) proposes a ranking loss to make Euclidean distance of feature close to a target $d$-dim sequence determined by the rank of tree metric. We work on the extension of Zeng et al. (2022) which is a special type of hierarchical methods where a regularizer is used. For a broader discussion on various hierarchical methods, we direct readers to the summary provided by Bertinetto et al. (2020).

**OT Methods and Hierarchical Learning**  The application of Optimal Transport (OT) in hierarchical learning contexts is explored in a limited number of studies. Ge et al. (2021), for instance, substitutes the conventional cross-entropy with a novel loss function that optimally transports the predictive distribution to the ground-truth one-hot vector, modifying the ground metric as tree metric and thus utilizing the hierarchical structure of the classes. Additionally, SEAL (Tan et al., 2023) employs the TWD to construct a latent hierarchy, enhancing traditional supervised and semi-supervised learning methodologies. It can be demonstrated that, given a tree $\mathcal{T}$, the optimization process for TWD aligns with $\ell_1$ loss optimization and is upper bounded by the SumLoss baseline introduced by Zeng et al. (2022). Consequently, these methodologies can be categorized as hierarchical loss functions rather than regularization methods discussed in this work.

## 6  CONCLUSION

Building upon Zeng et al. (2022) using CPCC as a regularizer to learn structured representations, we have identified and targeted the limitations inherent in $\ell_2$-CPCC. Our proposed OT-CPCC family provides a more nuanced measurement of pairwise class distances in feature space, effectively capturing complex distributions theoretically and empirically. Furthermore, our novel linear FastFT algorithm itself is interesting in nature for OT communities. FastFT-CPCC can be viewed as a successful application in learning tasks, and we welcome future work to investigate more on the properties of our greedy approach.

ACKNOWLEDGMENT

SZ and HZ are partially supported by an NSF IIS grant No. 2416897. HZ would like to thank the support from a Google Research Scholar Award. MY is partly supported by MEXT KAKENHI Grant Number 24K03004 and by JST ASPIRE JPMJAP2302. The views and conclusions expressed in this paper are solely those of the authors and do not necessarily reflect the official policies or positions of the supporting companies and government agencies.

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

APPENDIX

## A  APPROXIMATION METHODS OF EARTH MOVER'S DISTANCE

Most approximation methods do not rely on any tree structure. **Sinkhorn** (Cuturi, 2013) is a fully differentiable approximation method which leverages the entropic regularization setup of EMD and the convergence of doubly stochastic matrix. **Sliced Wasserstein Distance** (SWD) (Bonneel et al., 2015) projects data to 1 dimension through a random projection matrix and uses $\text{EMD}_{1d}$ for fast calculation, but the quality of approximation highly relies on the number of random projections. **Tree Wasserstein Distance** (TWD) (Yamada et al., 2022; Le et al., 2019; Takezawa et al., 2021) is a tree based approximation algorithm that can be written in the closed matrix form $\|\text{diag}(\text{w})\boldsymbol{B}(a - b)\|_1$, where w contains weights of $\mathcal{T}$ and $\boldsymbol{B}$ is a 0-1 internal nodes by leaf nodes matrix that represents if a node is the ancestor of a leaf node. To make sure TWD is a good approximation, an extra step of lasso regression is performed to learn w such that the tree metric approximates Euclidean distance between features with iterative gradient based method. In Yamada et al. (2022), it requires us to learn the tree distance between all pairs of nodes, which leads to quadratic computational complexity with respect to sample size as shown in Tab.1.

## B  COMPARISON OF STRUCTURED REPRESENTATION LEARNING AND LEARNING LABEL REPRESENTATIONS

Generally speaking, the problem of structured representation learning and the problem of learning label representations, or learning label embeddings, are two different topics. In label representation learning, each label is mapped to a high-dimensional vector for downstream tasks. For example, in standard multi-class classification, we typically use one-hot vector embeddings. Alternative approaches include using a soft label to assign probability values in (0,1) to each class (Nguyen et al., 2014), or learning label embeddings (Weston et al., 2011; Sun et al., 2017; Liu et al., 2024). Label representation learning methods have various downstream applications, such as text summarization (Sun et al., 2017), novelty detection (Barz & Denzler, 2019), and extreme classification (Jalan & Kar, 2019; Tagami, 2017; Bhatia et al., 2015; Yu et al., 2014; Evron et al., 2018; Gupta et al., 2019).

While $\ell_2$-CPCC falls into this category, OT-CPCC methods are not label embedding methods. Instead of representing a class with a Euclidean centroid in $\ell_2$, OT-CPCC methods intentionally avoid this step by considering a more nuanced relationship between labels. OT-CPCC relies on pairwise relationships between data points from two classes to include more complex distributional geometry during learning. Since these methods depend on data point relations, there is no one-to-one correspondence between a label and a representation.

## C  EMD-CPCC GENERALIZES $\ell_2$-CPCC

We can show EMD-CPCC is a generalization of previous work of $\ell_2$-CPCC through the following proposition.

**Proposition 3.1** (EMD reduces to $\ell_2$ between means of Gaussian with same covariance)**.** $\text{EMD}(\mathcal{N}(\mu_z, \Sigma), \mathcal{N}(\mu_{z'}, \Sigma)) = \|\mu_z - \mu_{z'}\|$.

*Proof.* Since $\mu, \nu$ in EMD definition are probability measures, the definition of EMD can be generalized to continuous distributions as well. Denote $Z$ the random variable from $\mu$, $Z'$ from $\nu$, and $q$ the coupling between $\mu$ and $\nu$, then EMD can be also expressed in the following $p$-Wasserstein ($\mathcal{W}_p$) form when $p = 1$:

$$
\begin{aligned}
\mathcal{W}_p &= \left(\inf_q \int_{\mathcal{Z} \times \mathcal{Z}} \|z_i - z_j\|^p\, q(z_i, z_j) dz_i z_j\right)^{\frac{1}{p}} \\
&= \inf_q (\mathbb{E}\|Z - Z'\|^p)^{\frac{1}{p}}
\end{aligned}
\tag{4}
$$

With this definition, our statement can be proved in two steps. First, we want to show the lower bound of EMD is $\|\mu_z - \mu_{z'}\|$ for any pairs of input random variables. By one step application of Jensen's

inequality, since every norm is convex, we have the following result from the expectation form of $\mathcal{W}_1$ in Eq.4:

$$\mathcal{W}_1 = \inf \mathbb{E} \, \|Z - Z'\| \geq \inf \|\mathbb{E}(Z - Z')\|$$
$$= \|\mathbb{E}Z - \mathbb{E}Z'\| = \|\mu_z - \mu_{z'}\| \tag{5}$$

Second, we need to show EMD $\leq \|\mu_z - \mu_{z'}\|$ for the given Gaussian constraints. Following the convexity of quadratic function, apply Jensen's again, we have:

$$\mathcal{W}_1^2 = (\inf \mathbb{E} \, \|Z - Z'\|)^2 \leq \inf \mathbb{E}(\|Z - Z'\|)^2 = \mathcal{W}_2^2 \tag{6}$$

Dowson & Landau (1982) showed the following analytical form of

$$\mathcal{W}_2^2 = \|\mu_z - \mu_{z'}\|^2 + \text{tr} \left( \Sigma_z + \Sigma_{z'} - 2(\Sigma_z \Sigma_{z'})^{1/2} \right) \tag{7}$$

This distance metric is also sometimes called **Bures-Wasserstein** (Bures) distance. The trace term becomes 0 if $\Sigma_z = \Sigma_{z'} = \Sigma$.

∎

## D    DETAILS FOR FAST FLOW TREE CPCC ALGORITHM

In this section, we first present the full pseudocode to compute FastFT-CPCC in Alg.3. Compared to FlowTree (Backurs et al., 2020) which uses the helper function in Alg.2 (Kalantari & Kalantari, 1995), we replace the `greedy_flow_matching` algorithm with Alg.1 which aligns with algorithm that computes the 1 dimensional OT transportation plan.

---

**Algorithm 3** Train with Fast FlowTree CPCC

---

**input** epoch $E$, batch size $B$, train data $\mathcal{D} = \{(x_i, y_i)\}_{i=1}^N$, dictionary $S$
1: **for** iteration $= 0, 1, \ldots, E - 1$ **do**
2:    **for** batch $= 0, 1, \ldots, \lfloor N/B \rfloor$ **do**
3:       assume there are $k$ classes in batch
4:       **for** $(a, b)$ **in** $\binom{k}{2}$ pairs of classes **do**
5:          $z_a = (a_1, \ldots, a_m)$
6:          $z_b = (b_1, \ldots, b_n)$
7:          **if** $(m, n)$ **in** $S$ **or** $(n, m)$ **in** $S$ **then**
8:             $P \leftarrow S[(m, n)]$
9:          **else**
10:             $P \leftarrow$ `greedy_flow_matching(m,n,a=unif,b=unif)`        {Alg.1}
11:             $S[(m, n)] \leftarrow P$
12:          idx $\leftarrow$ `P.nonzero()`                 {store nonzero index tuples of $P$}
13:          $\rho(a, b) = 0$
14:          **for** $(i, j)$ **in** idx **do**
15:             $\rho(a, b) \mathrel{+}= P[i, j] \cdot \text{norm}(z_a[i], z_b[j])$
16:       Calculate Eq.1 with CPCC as $\mathcal{R}$
17:       Update network parameters with backpropagation

---

Now we prove the correctness of Fast Flow Tree algorithm and its relation to 1 dimensional optimal transport distance.

**Theorem 3.2** (Correctness of Fast FlowTree). *For any augmented label tree $\mathcal{T}$, Alg. 1 and Alg. 2 return the same EMD approximation. The Fast FlowTree can be computed in $O((m + n)d)$.*

**Proof Sketch**   We can observe that the blue code blocks, which contain the update rule of the flow matrix, are almost identical between Alg.1 and Alg.2. According to Kalantari & Kalantari (1995), Alg.2 provides the optimal flow matrix for the optimal transport problem using the tree metric as the ground metric in $D$. From Fig.1, the tree distance between any two samples in the augmented tree is always the same, meaning the order of encountering any vertex within the source or target distribution does not matter. This effectively reduces the FlowTree problem in Alg.2 to a flat 1d EMD problem in Alg.1. The major difference between two algorithms is that, unlike Alg.2, Alg.1 does not require passing the flow from the leaf to the root node.

*Proof.* Alg.2 will not start to match leaves until the current node has children from two different distributions, or in our context, from two different fine labels. This is because of the while statement in line 8: based on our construction of $\mathcal{T}$ in Fig.2, if the current node $v$ is either a leaf or a fine label node, then all of its children are from either $C_a(v)$ or $C_b(v)$, so we will jump to line 11 to send all flows to $v$'s parent. Therefore, the matching starts at the lowest common ancestor of two label nodes.

Now we have $v$ as the lowest common ancestor of two fine class nodes (ex., $F$ or $H$ in Fig.2). Before entering line 9, we always pass unmatched demands to the parent, and we can see the demands are always from one label until we reach the lowest common ancestor. From Fig.2, we also know at this point, we have collected all flows from both labels. Therefore, the sum of flow is 1 for both fine label nodes when the matching starts.

Both Alg.2 and Alg.1 greedily remove one sample from either distribution. The greedy plan is always feasible and no unmatched flow is left after one scan of all $m + n$ features, because we inherently remove the the same amount of flow from both distributions every iteration in the while loop. Thus, on our $\mathcal{T}$, we can skip most steps in Alg.2: bottom up step in line 2, looping through all vertices at height $h$, the if-else check from line 4-7, and the last step of passing unmatched demands to parents are all unnecessary. Alg.2 terminates immediately once we run through all while loop steps, thus reducing to Alg.1.

Alg.1, or the flow matrix solution for 1-$d$ EMD problem, can be used when Monge Property or north-west-corner rules (Hoffman, 1963) hold. It can be shown by induction that this $O(m + n)$ greedy algorithm can be applied whenever the following constraint is satisfied: if $\{(z_i, a_i)\}, \{(z_j, b_j)\}$ are indexed (1d feature, weight) sequences, for indices $i_1 < i_2$ and $j_1 < j_2$:

$$\boldsymbol{D}_{i_1,j_1} + \boldsymbol{D}_{i_2,j_2} \leq \boldsymbol{D}_{i_1,j_2} + \boldsymbol{D}_{i_2,j_1} \tag{8}$$

If two sequences are ordered (in ascending order) by metric defining $\boldsymbol{D}$, we have $z_{i_1} \leq z_{i_2}$ and $z_{j_1} \leq z_{j_2}$. There are in total six cases for any $i_1 < i_2, j_1 < j_2$:

1. If $z_{j_1} \leq z_{j_2} \leq z_{i_1} \leq z_{i_2}$, then Eq.8 $\Leftrightarrow z_{i_1} - z_{j_1} + z_{i_2} - z_{j_2} \leq z_{i_1} - z_{j_2} + z_{i_2} - z_{j_1}$ holds.
2. If $z_{j_1} \leq z_{i_1} \leq z_{j_2} \leq z_{i_2}$, then Eq.8 $\Leftrightarrow z_{i_1} - z_{j_1} + z_{i_2} - z_{j_2} \leq z_{j_2} - z_{i_1} + z_{i_2} - z_{j_1}$ holds because $z_{i_1} \leq z_{j_2}$.
3. If $z_{j_1} \leq z_{i_1} \leq z_{i_2} \leq z_{j_2}$, then Eq.8 $\Leftrightarrow z_{i_1} - z_{j_1} + z_{j_2} - z_{i_2} \leq z_{j_2} - z_{i_1} + z_{i_2} - z_{j_1}$ holds because $z_{i_1} \leq z_{i_2}$.
4. If $z_{i_1} \leq z_{j_1} \leq z_{j_2} \leq z_{i_2}$, then Eq.8 $\Leftrightarrow z_{j_1} - z_{i_1} + z_{j_2} - z_{i_2} \leq z_{j_2} - z_{i_1} + z_{i_2} - z_{j_1}$ holds because $z_{j_1} \leq z_{i_2}$.
5. If $z_{i_1} \leq z_{j_1} \leq z_{i_2} \leq z_{j_2}$, then Eq.8 $\Leftrightarrow z_{j_1} - z_{i_1} + z_{j_2} - z_{i_2} \leq z_{j_2} - z_{i_1} + z_{i_2} - z_{j_1}$ holds because $z_{j_1} \leq z_{i_2}$.
6. If $z_{i_1} \leq z_{i_2} \leq z_{j_1} \leq z_{j_2}$, then Eq.8 $\Leftrightarrow z_{j_1} - z_{i_1} + z_{j_2} - z_{i_2} \leq z_{j_2} - z_{i_1} + z_{j_1} - z_{i_2}$ holds.

Therefore, if two sequences are ordered with respect to some metric, Eq.8 is satisfied.

Recall we need the extra sorting step in the beginning of 1d OT plan to apply the greedy flow matching algorithm. The distance function in Eq.8 is decided by the ground metric of EMD, which does not make any difference for 1 dimensional features. In Flow Tree, a tree is constructed with QuadTree such that the tree distance of any two features is close to the $\ell_2$ distance between them. In this case, in EMD computation, we can improve computational efficiency by Flow Tree. With extended label tree $\mathcal{T}$, because all same class sample leaves belong to one single parent, we have a special case here where the tree distance of any two samples from $\mu, \nu$ are always the same, satisfying Eq.8 vacuously. This statement fails only if we assign different edge weights for each sample during the extension of label tree, and does not depend on the weights of original label tree. To avoid the limitation on the extended edge weights, we can instead assign different weights $a_i$ for each sample to adjust the importance of each dat points. Therefore, FastFT is still very flexible.

For the time complexity, Alg.1 requires at most $O(m+n)$ steps from line 4. The most time-consuming aspect is computing the $d$-dimensional distance of $O(m + n)$ pairs in $\boldsymbol{D}$, since there will be at most $m + n$ nonzero values in $\boldsymbol{D}$ after applying the greedy matching algorithm, which makes the final time complexity as $O((m + n)d)$. ∎

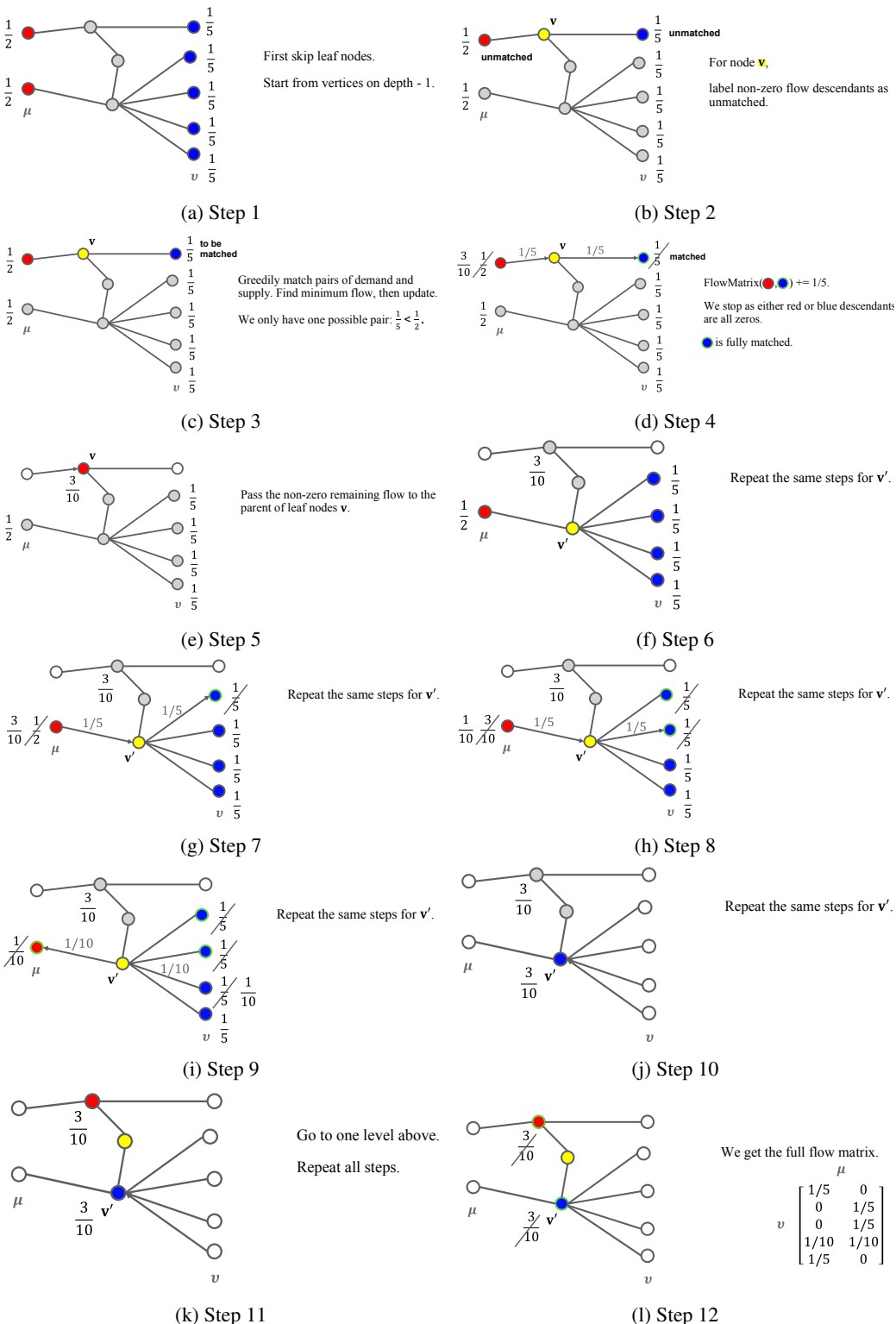

Figure 8: Example run of Alg.2 Bottom-up Tree Flow Matching.

# E  DATASET HIERARCHY CONSTRUCTION

In this section, we discuss the details of dataset hierarchies and how we created the dataset splits. Throughout the main body of this paper, we consider a two-level hierarchy: coarse and fine. For each coarse node, except for *BREEDS*, we split the coarse node's fine children into approximately 40% for the target set and 60% for the source set. Following this procedure, the splits are detailed in Tables 6, 7, 8, 9, 10, 11, and 12. The original class labels for CIFAR10 and CIFAR100 are from https://www.cs.toronto.edu/~kriz/cifar.html, for iNaturalist from https://github.com/visipedia/inat_comp/tree/master/2021, and for BREEDS from WordNet (Miller, 1995).

The CIFAR10 hierarchy was manually constructed by us, while the CIFAR100 hierarchy is provided in the original dataset. In INAT, fine and coarse classes correspond to the `class` and `phylum` labels from the original annotations. These higher-level labels are particularly suitable for evaluating multi-modal settings, where OT-CPCC methods may have a more significant advantage over $\ell_2$-CPCC.

In Zeng et al. (2022), fine classes in BREEDS are defined one level deeper than the original labels. However, due to the large number of classes and limited samples per class within a batch, the distinction between $\ell_2$-CPCC and OT-CPCC can become less apparent. To address this, we employ a three-level hierarchy, using the labels from Santurkar et al. (2021) as fine labels and setting their parent categories as mid and coarse labels. Consequently, due to BREEDS setting, the fine classes are the same for both source and target datasets, so there is no source-target split in Tables 9, 10, 11, and 12. Despite this hierarchical structure, as seen in Fig.5, BREEDS exhibits relatively low multi-modality. With this design, in Sec.4.3, for classification and retrieval experiments, one-shot fine-tuning is not required for BREEDS on the target split.

Table 6: CIFAR10 hierarchy and data split.

| Coarse | Source | Target |
|---|---|---|
| animal | deer, dog, frog | bird, cat |
| transportation | ship, truck | airplane, automobile |

Table 7: CIFAR100 hierarchy and data split.

| Coarse | Source | Target |
|---|---|---|
| aquatic mammals | otter, seal, whale | beaver, dolphin |
| fish | ray, shark, trout | aquarium fish, flat fish |
| flowers | roses, sunflowers, tulips | orchids, poppies |
| food containers | cans, cups, plates | bottles, bowls |
| fruit and vegetables | oranges, pears, sweet pepper | apples, mushrooms |
| household electrical devices | lamp, telephone, television | clock, computer keyboard |
| household furniture | couch, table, wardrobe | bed, chair |
| insects | butterfly, caterpillar, cockroach | bee, beetle |
| large carnivores | lion, tiger, wolf | bear, leopard |
| large man-made outdoor things | house, road, skyscraper | bridge, castle |
| large natural outdoor scenes | mountain, plain, sea | cloud, forest |
| large omnivores and herbivores | chimpanzee, elephant, kangaroo | camel, cattle |
| medium-sized mammals | possum, raccoon, skunk | fox, porcupine |
| non-insect invertebrates | snail, spider, worm | crab, lobster |
| people | girl, man, woman | baby, boy |
| reptiles | lizard, snake, turtle | crocodile, dinosaur |
| small mammals | rabbit, shrew, squirrel | hamster, mouse |
| trees | palm, pine, willow | maple, oak |
| vehicles 1 | motorcycle, pickup truck, train | bicycle, bus |
| vehicles 2 | streetcar, tank, tractor | lawn-mower, rocket |

Table 8: INAT hierarchy and data split.

| Coarse | Source | Target |
|---|---|---|
| Arthropoda | Diplopoda, Hexanauplia, Insecta, Malacostraca, Merostomata | Arachnida, Chilopoda |
| Chordata | Ascidiacea, Aves, Elasmobranchii, Mammalia, Reptilia | Actinopterygii, Amphibia |
| Cnidaria | Hydrozoa, Scyphozoa | Anthozoa |
| Echinodermata | Echinoidea, Holothuroidea, Ophiuroidea | Asteroidea |
| Mollusca | Cephalopoda, Gastropoda, Polyplacophora | Bivalvia |
| Ascomycota | Lecanoromycetes, Leotiomycetes, Pezizomycetes, Sordariomycetes | Arthoniomycetes, Dothideomycetes |
| Basidiomycota | Dacrymycetes, Pucciniomycetes, Tremellomycetes | Agaricomycetes |
| Bryophyta | Polytrichopsida, Sphagnopsida | Bryopsida |
| Tracheophyta | Liliopsida, Lycopodiopsida, Magnoliopsida, Pinopsida, Polypodiopsida | Cycadopsida, Gnetopsida |

Table 9: LIVING17 dataset hierarchy.

| Coarse | Fine |
|---|---|
| bird | grouse, parrot |
| amphibian | salamander |
| reptile, reptilian | turtle, lizard, snake, serpent, ophidian |
| arthropod | spider, crab, beetle, butterfly |
| mammal, mammalian | dog, domestic dog, Canis familiaris, wolf, fox, domestic cat, house cat, Felis domesticus, Felis catus, bear, ape, monkey |

Table 10: NONLIVING26 dataset hierarchy.

| Coarse | Fine |
|---|---|
| structure, place | dwelling, home, domicile, abode, habitation, dwelling house, fence, fencing, mercantile establishment, retail store, sales outlet, outlet, outbuilding, roof |
| paraphernalia | ball, bottle, digital computer, keyboard instrument, percussion instrument, percussive instrument, pot, stringed instrument, timepiece, timekeeper, horologe, wind instrument, wind |
| food, nutrient | squash |
| apparel, toiletries | bag, body armor, body armour, suit of armor, suit of armour, coat of mail, cataphract, coat, hat, chapeau, lid, skirt |
| conveyance, transport | boat, bus, autobus, coach, charabanc, double-decker, jitney, motorbus, motorcoach, omnibus, passenger vehicle, car, auto, automobile, machine, motorcar, ship, truck, motortruck |
| furnishing | chair |

## F  TRAINING HYPERPARAMETERS

We conducted all experiments with NVIDIA RTX A6000. The architectural and training setups for each dataset are as follows:

Table 11: ENTITY13 dataset hierarchy.

| Coarse | Fine |
|---|---|
| living thing, animate thing non-living thing | bird, reptile, reptilian, arthropod, mammal, mammalian garment, accessory, accoutrement, accouterment, craft, equipment, furniture, piece of furniture, article of furniture, instrument, man-made structure, construction, wheeled vehicle, produce, green goods, green groceries, garden truck |

Table 12: ENTITY30 dataset hierarchy.

| Coarse | Fine |
|---|---|
| living thing, animate thing | serpentes, passerine, passeriform bird, saurian, arachnid, arachnoid, aquatic bird, crustacean, carnivore, insect, ungulate, hoofed mammal, primate, bony fish |
| non-living thing | barrier, building, edifice, electronic equipment, footwear, legwear, garment, headdress, headgear, home appliance, household appliance, kitchen utensil, measuring instrument, measuring system, measuring device, motor vehicle, automotive vehicle, musical instrument, instrument, neckwear, sports equipment, tableware, tool, vessel, watercraft, dish, vegetable, veggie, veg, fruit |

## F.1 CIFAR10, CIFAR100

We conduct training from scratch using ResNet18, following the ResNet architecture design where the first convolution layer employs a 3x3 kernel size, while the max pooling layer is omitted. The images are preprocessed by normalizing them according to CIFAR10/CIFAR100's mean and standard deviation. We augment the training data by first zero-padding the images by 4 pixels, then cropping them back to their original dimensions, applying a 50% chance of horizontal flipping, and randomly rotating the images between -15 and 15 degrees.

For CIFAR10, the pretraining process spans 100 epochs with a batch size of 64. The learning rate begins at 0.01 and is decreased by a factor of 10 every 60 steps, with momentum set at 0.9 and weight decay set to $5 \times 10^{-4}$ in the SGD optimizer. For CIFAR100, the pretraining process spans 200 epochs with a batch size of 128. The learning rate begins at 0.1 and is decreased by a factor of 5 every 60 steps, with momentum set at 0.9 and weight decay set to $5 \times 10^{-4}$ in the SGD optimizer. Additionally, for both datasets, in the fine-level one-shot transfer learning, we train for 100 epochs starting at a learning rate of 0.1, which is reduced by a factor of 10 every 60 steps. CPCC is applied to coarse, fine level and $\lambda$ in Eq.1 is set to 1. The ratio between the distance between two fine classes sharing the same coarse parent and those sharing different parents is 1:2.

## F.2 BREEDS

We train a ResNet18 model from scratch originally designed for ImageNet, where the first convolution layer uses a 7x7 kernel. Adhering to the BREEDS training setup, we use a batch size of 128, an initial learning rate of 0.1, and SGD optimization with a weight decay of $10^{-4}$. For the *ENTITY13* and *ENTITY30* datasets, the model is trained for 300 epochs, reducing the learning rate by a factor of 10 every 100 steps, while for *LIVING17* and *NONLIVING26*, we extend training to 450 epochs, with the learning rate divided by 10 every 150 steps. Data augmentation for the training set includes a RandomResizedCrop to 224x224, horizontal flipping with a 50% probability, and color jittering that adjusts brightness, contrast, and saturation within a range of 0.9 to 1.1. The test set, on the other hand, is resized to 256x256 and then center-cropped to 224x224.

Since in our experiments with BREEDS, the source and target split share the same set of fine labels, to calculate fine target accuracy, it is unnecessary to use one-shot transfer learning. We directly use the source set pretrained models for target fine accuracy evaluation.

To apply CPCC methods, we backtrack two levels up to find the mid and coarse labels from the original class labels in BREEDS. CPCC is applied to mid and coarse level and $\lambda$ in Eq.1 is set to 1. The ratio between the distance between two fine classes sharing the same coarse parent and those sharing different parents is 1:2.

## F.3 INAT

We fine-tune an ImageNet-pretrained ResNet18 model where checkpoint can be downloaded in Pytorch (`torchvision.models.resnet18(weights='IMAGENET1K_V1')`, `https://pytorch.org/vision/main/models/generated/torchvision.models.resnet18.html`). The images are preprocessed by normalizing them with mean=[0.466, 0.471, 0.380] and standard deviation=[0.195, 0.194, 0.192]. We augment the training data by cropping a random portion (scale=(0.08, 1.0), ratio=(0.75, 1.33)) of image and resize it to 224x224, and apply a 50% chance of horizontal flipping. We use a batch size of 1024. The learning rate begins at 0.1 and is decreased by a factor of 5 every 60 steps, with momentum set at 0.9 and weight decay set to $5 \times 10^{-4}$ in the SGD optimizer. In the downstream one-shot fine-tuning task, we train for 60 epochs starting at a learning rate of 0.05, which is reduced by a factor of 10 every 15 steps. CPCC is applied to coarse, fine level and $\lambda$ in Eq.1 is set to 0.1. The ratio between the distance between two fine classes sharing the same coarse parent and those sharing different parents is 1:3.

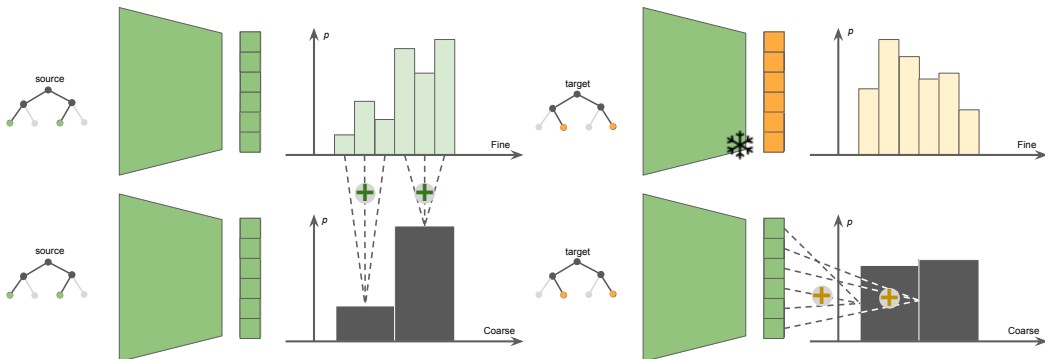

Figure 9: Visualization of the evaluation of hierarchical classification.

## G    HIERARCHICAL BASELINE OBJECTIVES

In this section, we introduce the details of hierarchical objectives we used in Fig.12.

**Flat**    is the naive cross entropy loss only on the fine level, so the expression is:

$$\mathcal{L}_{\text{Flat}}(x, y_{\text{fine}}) = \sum_{(x,y) \in \mathcal{D}} \ell_{\text{Flat}}(y_{\text{fine}}, h(x)). \tag{9}$$

We adopt all hierarchical baselines from the comparative study in Zeng et al. (2022), including:

**Multi-Task Learning** (MTL) (Caruana, 1997), where the backbone model is trained with two linear heads—one for coarse classes and one for fine classes—each with a cross-entropy loss. The total loss, $\mathcal{L}_{\text{MTL}}$, is the sum of these two cross-entropy terms, with a scaling coefficient of 1 for both. **Curriculum Learning** (Curr) (Bengio et al., 2009) involves two training phases: first, training a coarse head for 50 epochs, followed by training the full network with a fine head for 200 epochs, starting from the coarse-trained features, assuming that coarse classification is an easier task. **Sum Loss** (Hu et al., 2018) uses only one fine classifier head, where coarse predictions are obtained by summing the fine class probabilities for the same coarse parent and applying a second cross-entropy loss. **Hierarchical Cross Entropy** (HXE) (Bertinetto et al., 2020) is the sum of weighted cross-entropy losses ($\alpha = 0.4$), where the hierarchical structure is incorporated into the log-probabilities. The same paper also introduces **Soft Label** (Soft), which uses a single softmax loss based on the height of the least common ancestor between label pairs, with a hyperparameter $\beta = 10$. **Quadruplet**

**Loss** (Quad) (Zhang et al., 2016) extends triplet loss by defining two types of positive labels based on the hierarchy. For detailed mathematical expressions of these objectives, refer to App.C in Zeng et al. (2022).

We also include several recently published hierarchical losses:

**Soft Hierarchical Consistency** (SHC) (Garg et al., 2022) is the combination of MTL and Sum Loss. For a two level hierarchy, they jointly train a two-headed network. Define the network as $h : \mathcal{X} \rightarrow \Delta_{k_1} \times \Delta_{k_2}$ with a shared feature encoder $f : \mathcal{X} \rightarrow \mathcal{Z}$ and two classifiers $g_{\text{coarse}} : \mathcal{Z} \rightarrow \Delta_{k_1}, g_{\text{fine}} : \mathcal{Z} \rightarrow \Delta_{k_2}$, with $k_1$ as the number of coarse classes and $k_2$ the number of fine classes. The ground truth coarse probability $p_{\text{coarse}} = g_{\text{coarse}}(f(\cdot))$ is the output of coarse head after softmax normalization, and $\hat{p}_{\text{coarse}}$ is derived from the output of fine head. Let $\mathbf{W}$ be a $k_1$ by $k_2$ matrix representing the relationships in the label tree: if a fine class $i$ belongs to a coarse class $j$, then $\mathbf{W}_{ji}$ is 1, else the entry is set to 0. Then $\hat{p}_{\text{coarse}} = \mathbf{W} g_{\text{fine}}(f(\cdot))$, so the probability of belonging to a given coarse class is the sum of the probabilities of its fine class descendants in the tree. Then, the SHC loss is defined as the Jensen-Shannon Divergence between these two probability distributions:

$$\mathcal{L}_{\text{SHC}}(x, y_{\text{coarse}}, y_{\text{fine}}) = \text{JSD}(p_{\text{coarse}}(x, y_{\text{coarse}}) || \hat{p}_{\text{coarse}}(x, y_{\text{fine}})). \tag{10}$$

In Garg et al. (2022), $\mathcal{L}_{\text{SHC}}$ is used in the combination with Flat cross entropy loss. Therefore, we treat it as a counterpart of CPCC losses and use $\mathcal{L}_{\text{Flat}} + \lambda \cdot \mathcal{L}_{\text{SHC}}$ as the training loss, with $\lambda = 1$.

**Geometric Consistency** (GC) (Garg et al., 2022) is also computed under the MTL two-head pipeline. The coarse classifier can be decomposed to a linear layer and softmax function, i.e., let $\mathbf{W}^c \in \mathbb{R}^{d \times k_1}$, $g_{\text{coarse}}(\cdot) = \text{softmax}(\mathbf{W}^c(\cdot))$, and we define $\mathbf{W}^f$ similarly for the linear fine layer.

$$\mathcal{L}_{\text{GC}}(x, y_{\text{coarse}}, y_{\text{fine}}) = \sum_{i=1}^{k_1} 1 - \cos(\mathbf{W}_i^c, \hat{\mathbf{W}}_i^c). \tag{11}$$

As in SHC, GC aims to force the similarity between the predicted coarse weight and the true coarse weight. Here, $\mathbf{W}_i^c$ is part of the linear classifier corresponds to each coarse class, and the predicted coarse weight $\hat{\mathbf{W}}_i^c = \sum_{j \in \text{children}(i)} \mathbf{W}_j^f / \left\| \sum_{j \in \text{children}(i)} \mathbf{W}_j^f \right\|$ is the normalized sum of a part of fine classifiers based on the hierarchical relationship. We combine $\mathcal{L}_{\text{GC}}$ with Flat cross entropy during training and set $\lambda$ in Eq.1 as 1.

**RankLoss** (Rank) (Nolasco & Stowell, 2022) proposes a ranking loss to make Euclidean distance of feature $\rho$ close to the target $d$-dim sequence determined by the rank of tree metric:

$$\mathcal{L}_{\text{rank}}(x, y_{\text{coarse}}, y_{\text{fine}}) = \frac{1}{p} \sum_p (1 - I_p)(\rho - \text{TargetRank}(\rho)). \tag{12}$$

In this equation above, $I_p$ is an indicator variable which shows whether the rank of pairwise $\ell_2$ feature distance is the same as the rank of pairwise tree metric distance, and TargetRank is a function reflecting hierarchical similarity. Since this loss does not focus on discriminative tasks like classification, we combine it with Flat cross entropy, setting $\lambda = 1$ in Eq.1 for downstream tasks evaluation.

## H  ADDITIONAL EXPERIMENTS

### H.1  EFFECT OF TREE DEPTH

Although in the main body we focus on a depth-3 label hierarchy consisting of a root node, a coarse level, and a fine level, we also test whether our method can be applied to more complex trees. In CIFAR100, beyond the standard fine-coarse levels, we introduce a mid level by grouping 5 fine classes from the same coarse class into two smaller groups (2 + 3). Additionally, we create a coarser level by grouping two alphabetically neighboring coarse classes into a single coarser label. For example, for 10 fine labels, the hierarchical grouping is highlighted by the following color scheme: {[(orchids, poppies) (roses, sunflowers, tulips)] [(bottles, bowls) (cans, cups, plates)]}.

In Fig.10, we aligned the axes to sort the fine labels based on the hierarchical structure for embeddings learned using trees of varying depth. We then computed the fine-to-fine class distances using either

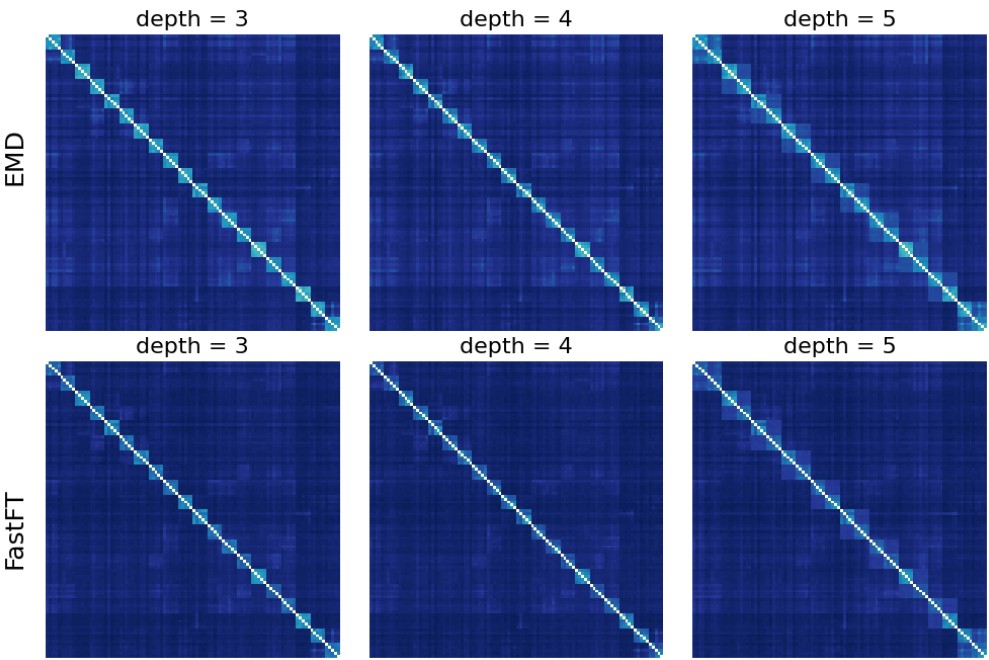

Figure 10: CIFAR100 class embedding pairwise distance matrix for different types of label tree with varying tree depth.

the EMD or FastFT metric. On the left of Fig.10, we used the standard fine-coarse CIFAR100 label tree, where a clear pattern of twenty 5x5 coarse blocks appears on the diagonal for both optimal transport methods. In the middle of Fig.10, we applied CPCC to the fine-mid-coarse hierarchy, and within each 5x5 coarse block, we observed the expected fine-grained 2x2 + 3x3 patterns. Similarly, on the right of Fig.10, when we included the coarser grouping, we could observe the merging of neighboring 5x5 coarse blocks.

Table 13: EMD and FastFT performance for tree with different depths on the full CIFAR100 dataset. The batch size is set to 128.

| Objective | Setup | CPCC | FineAcc | CoarseAcc |
|---|---|---|---|---|
| EMD | depth = 3 | 88.80 | 71.91 | 82.21 |
| | depth = 4 | 85.09 | 72.59 | 82.93 |
| | depth = 5 | 88.64 | 72.20 | 82.50 |
| FastFT | depth = 3 | 90.75 | 72.71 | 83.31 |
| | depth = 4 | 87.86 | 72.45 | 83.05 |
| | depth = 5 | 90.27 | 72.56 | 83.04 |

We also include metrics for the preciseness of hierarchical information and hierarchical classification in Tab.13. As observed in Fig.10, the tree information is well-learned, leading us to expect high CPCC scores across all settings in Tab.13. For the hierarchical classification metrics, within the same OT method, the difference between FineAcc and CoarseAcc is small. It is worth noting that, because we train on the full CIFAR100 dataset for visualization purposes, the source-target hierarchical split shown in Fig.4 is not applied here. Both FineAcc and CoarseAcc are reported on the original CIFAR100 test set. This suggests that even when the tree used is not identical to the label hierarchy associated with the dataset, an approximate tree with a similar structure is sufficient for classification tasks.

In summary, EMD-CPCC is a flexible structured representation learning method that adapts well to complex hierarchies, while FastFT-CPCC serves as a good approximation to EMD-CPCC. This

observation is consistent with the results in Tab.3, where optimal transport methods achieve higher Test CPCC values.

## H.2 EFFECT OF TREE WEIGHTS

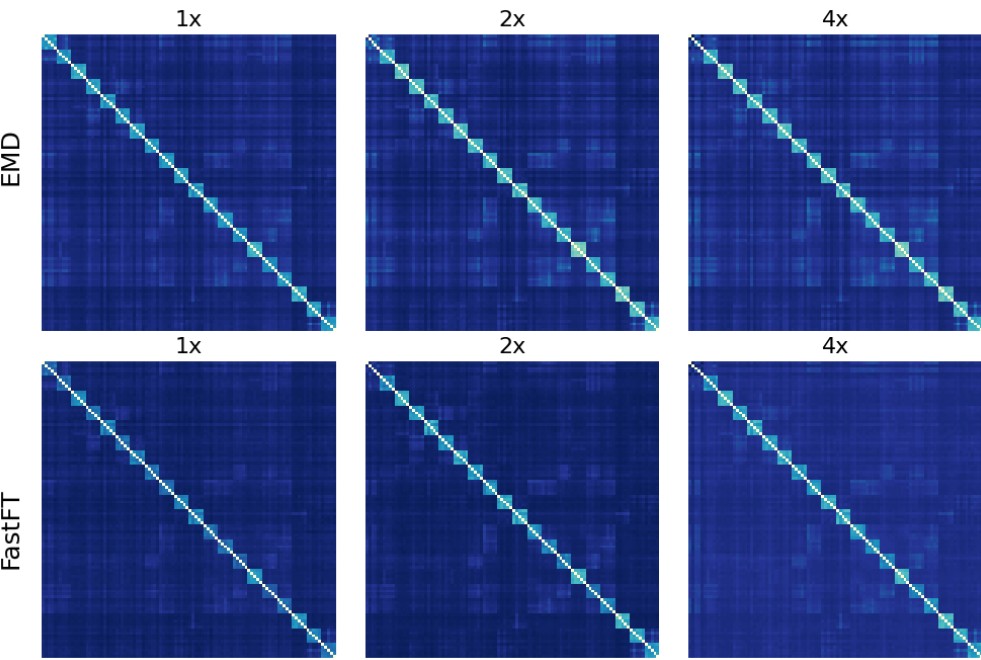

Figure 11: CIFAR100 class embedding pairwise distance matrix for different types of label tree with varying tree weights.

In this section, we demonstrate that all CPCC methods, including OT-CPCC, are influenced by input tree weights. In Fig.11, we begin with the original unweighted CIFAR100 tree (where all edges have a default weight of 1) and modify the edge weights of a subtree representing the coarse label `aquatic_mammals`, which includes five fine labels: `beaver`, `dolphin`, `otter`, `seal`, and `whale`. We visualize the learned representation through a distance matrix, as described in Sec.H.1. From left to right in Fig.11, the edge weights of this subtree are set to 1, 2, and 4, respectively. Consequently, we observe a color shift in the top-left coarse block of the matrix, with the block gradually darkening as the subtree weights increase. This demonstrates the flexibility of OT-CPCC methods, which can successfully embed weighted trees, and highlights why CPCC is a sensitive and effective metric for evaluating the interpretability and quality of the hierarchical information embedded in the learned representation, as discussed in Sec.4.4.

Table 14: EMD and FastFT performance for tree with different edge weights within `aquatic_mammals` classes on the full CIFAR100 dataset. The batch size is set to 1024.

| Objective | Setup | CPCC | FineAcc | CoarseAcc |
|-----------|-------|------|---------|-----------|
| EMD | weight 1x | 89.13 | 75.76 | 85.50 |
| | weight 2x | 88.85 | 75.57 | 85.22 |
| | weight 4x | 79.86 | 75.56 | 85.35 |
| FastFT | weight 1x | 92.12 | 75.43 | 85.43 |
| | weight 2x | 92.57 | 75.38 | 85.10 |
| | weight 4x | 84.23 | 75.57 | 84.98 |

We use the same experimental settings as those described in Sec.H.1. While the classification metrics remain comparable, the most significant trend emerges in the CPCC metric. As shown in

Fig.11, we observe the expected pattern in the top-left block, which changes as the edge weights are multiplied. However, as the injected hierarchy deviates further from the given label hierarchy, it becomes increasingly unnatural. The increased weights make the tree embedding more challenging, resulting in a drop in CPCC values for larger weights.

## H.3 EFFECT OF FLOW WEIGHT

In Eq.3, the flow weight vectors $a$ and $b$ are set to uniform by default in all other experiments. However, in practice, these vectors can be customized by the user if prior information about the data distribution is available, which may significantly influence the learned embeddings. On CIFAR100, we tested three different flow weight settings:

- **Uniform** (unif): default setting where we assume each data point is uniformly sampled from the class distribution, so $a = (1/m, \ldots, 1/m)$ and $b = (1/n, \ldots, 1/n)$.

- **Distance** (dist): is an adaptive weighting method where during training, we compute the Euclidean centroid of each class and compute the distance of each sample to the corresponding class centroid. Then we can apply softmax function to normalize the distances into probabilities. Mathematically, if two class representations are $Z, Z'$, then for each entry of flow weight vector, $a_i = \frac{\exp(Z_i - \mu_z)}{\sum_{j=1}^m \exp(Z_j - \mu_z)}$, $b_i = \frac{\exp(Z'_i - \mu_{z'})}{\sum_{j=1}^n \exp(Z'_j - \mu_{z'})}$.

- **Inverse-Distance** (inv): is similar to dist, yet the only difference is we set the weight to be inversely scaled by the distance to the centroid, so the expressions of the flow weights are $a_i = \frac{\exp(\mu_z - Z_i)}{\sum_{j=1}^m \exp(\mu_z - Z_j)}$, $b_i = \frac{\exp(\mu_{z'} - Z'_i)}{\sum_{j=1}^n \exp(\mu_{z'} - Z'_j)}$.

From Tab.15, we observe two key findings: first, tuning the flow weights can lead to improvements in out-of-distribution generalization metrics for the target set, but this comes at the expense of in-distribution source metrics. Second, the most significant differences are seen in the CPCC values, where it is nearly impossible to recover hierarchical information on the test set when using the Distance flow weight for both EMD and FastFT. The low interpretability scores suggest that not all flow weight settings are suitable for hierarchy learning.

Table 15: Comparison of FastFT and EMD methods with different optimal transport flow weighting schemes across fine and coarse metrics, with cell colors representing the magnitude of values.

| Objective | Weight | CPCC | Fine Metrics | | | Coarse Metrics | | | |
|---|---|---|---|---|---|---|---|---|---|
| | | | sAcc | sMAP | tAcc | sAcc | sMAP | tAcc | tMAP |
| FastFT | unif | 93.85 | 81.32 | 78.88 | 23.15 | 86.96 | 91.71 | 44.29 | 42.20 |
| | dist | 9.30 | 78.90 | 67.33 | 25.28 | 85.18 | 91.47 | 45.00 | 43.78 |
| | inv | 66.52 | 79.73 | 68.31 | 25.45 | 85.81 | 92.03 | 43.70 | 43.70 |
| EMD | unif | 89.60 | 80.99 | 78.51 | 23.53 | 86.93 | 91.82 | 44.03 | 42.32 |
| | dist | 19.87 | 79.23 | 73.79 | 26.05 | 85.91 | 91.30 | 43.55 | 42.53 |
| | inv | 98.13 | 79.51 | 75.84 | 26.80 | 86.05 | 91.22 | 44.15 | 42.13 |

## I ANALYSIS FOR THE TIME COMPLEXITY OF OT APPROXIMATION METHODS

In this section, we provide the details and references for the reported time complexities in Tab.1.

In the following time complexity analysis, denote $b$ as batch size. We assume each class-conditioned cluster has the same size $n$, which implies $b = nk$, where $k$ is the number of classes.

For $\ell_2$, given one batch of batch size $b$, computing Euclidean centroids for all classes requires scanning through all $d$-dimensional data points once, which requires $O(bd) = O(nkd)$. We have at most $k$ classes in one batch, so computing the pairwise distances between class centroids will cost $O(k^2 d)$. The total cost for $\ell_2$ is $O((nk + k^2)d)$.

Table 16: Detailed generalization performance on fine level. Each value is the mean (std) of three seeds of experiments. Flat directly optimizes on the fine level classes, so we treat it as the upper bound of fine metrics and colored in grey. The bold values have the largest mean for a column metric within a dataset, ignoring the upper bound.

| Dataset | Objective | sAcc | tAcc | sMAP | Dataset | sAcc | tAcc | sMAP |
|---|---|---|---|---|---|---|---|---|
| CIFAR10 | Flat | 96.92 (0.14) | 54.23 (6.92) | 99.43 (0.02) | INAT | 88.70 (0.01) | 26.77 (0.04) | 63.97 (0.04) |
| | $\ell_2$ | 96.96 (0.16) | 55.71 (7.38) | 99.22 (0.06) | | 88.62 (0.01) | 26.78 (0.04) | 56.10 (0.05) |
| | FastFT | 96.90 (0.13) | 55.99 (8.19) | 99.24 (0.02) | | 88.49 (0.01) | **27.10 (0.04)** | 56.21 (0.05) |
| | EMD | **97.05 (0.18)** | 56.12 (7.63) | 99.24 (0.02) | | **88.68 (0.00)** | 26.78 (0.04) | **56.83 (0.00)** |
| | Sinkhorn | 96.95 (0.08) | 54.89 (8.59) | 99.27 (0.03) | | 88.08 (0.00) | 26.77 (0.04) | 51.56 (0.04) |
| | SWD | 96.96 (0.05) | **59.21 (4.42)** | **99.45 (0.03)** | | 88.46 (0.00) | 26.78 (0.04) | 54.19 (0.03) |
| CIFAR100 | Flat | 80.53 (0.31) | 28.44 (1.89) | 86.47 (0.21) | BREEDS | 82.08 (0.26) | 45.19 (0.38) | 73.74 (0.46) |
| | $\ell_2$ | 80.98 (0.17) | 23.76 (1.01) | 77.52 (0.12) | | 82.66 (0.41) | 45.95 (0.40) | 62.86 (0.38) |
| | FastFT | 81.09 (0.18) | 24.71 (0.90) | 78.72 (0.19) | | 82.58 (0.34) | 45.75 (0.45) | 63.95 (0.36) |
| | EMD | **81.32 (0.19)** | 23.15 (0.40) | 78.88 (0.18) | | - | - | - |
| | Sinkhorn | 80.99 (0.12) | 23.53 (0.48) | 78.51 (0.20) | | **82.99 (0.31)** | **46.87 (0.40)** | 61.54 (0.55) |
| | SWD | 80.50 (0.33) | **26.18 (1.04)** | **86.82 (0.22)** | | 82.90 (0.30) | 46.33 (0.16) | **76.24 (0.39)** |

Table 17: Detailed generalization performance on coarse level. Each value is the mean (std) of three seeds of experiments. $\ell_2$ directly optimizes on the coarse level classes, so we treat it as the upper bound of coarse metrics and colored in grey.

| Dataset | Objective | sAcc | tAcc | sMAP | tMAP | Dataset | sAcc | tAcc | sMAP | tMAP |
|---|---|---|---|---|---|---|---|---|---|---|
| CIFAR10 | $\ell_2$ | 99.60 (0.01) | 87.51 (0.97) | 99.88 (0.03) | 92.84 (0.63) | INAT | 94.52 (0.00) | 40.86 (0.01) | 74.34 (0.05) | 37.35 (0.01) |
| | Flat | 99.58 (0.07) | 87.30 (0.59) | 99.22 (0.05) | 89.66 (0.38) | | 94.63 (0.00) | 38.81 (0.01) | 70.41 (0.04) | 34.00 (0.13) |
| | FastFT | 99.61 (0.06) | **87.79 (0.41)** | **99.91 (0.03)** | 93.04 (0.57) | | **94.66 (0.00)** | 39.43 (0.02) | 72.90 (0.05) | 35.63 (0.01) |
| | EMD | **99.61 (0.03)** | 87.45 (0.60) | 99.88 (0.03) | **93.07 (0.63)** | | 94.64 (0.00) | **41.01 (0.01)** | 73.87 (0.01) | 35.58 (0.01) |
| | Sinkhorn | 99.61 (0.05) | 87.41 (0.53) | 99.87 (0.03) | 92.60 (0.13) | | 94.30 (0.00) | 36.94 (0.02) | 68.75 (0.02) | 34.92 (0.00) |
| | SWD | 99.56 (0.04) | 87.63 (0.62) | 99.36 (0.09) | 90.12 (0.43) | | 94.52 (0.00) | 39.38 (0.01) | **75.13 (0.03)** | **38.11 (0.01)** |
| CIFAR100 | $\ell_2$ | 87.00 (0.13) | 44.55 (0.39) | 92.13 (0.23) | 43.03 (0.55) | BREEDS | 94.24 (0.30) | 83.13 (0.23) | 95.00 (0.25) | 81.73 (0.32) |
| | Flat | 86.17 (0.26) | 43.09 (0.55) | 82.07 (0.34) | 42.09 (0.26) | | 93.85 (0.20) | 82.43 (0.27) | 74.47 (0.52) | 66.50 (0.39) |
| | FastFT | **86.96 (0.13)** | **44.29 (0.33)** | 91.71 (0.11) | 42.20 (0.44) | | 94.25 (0.27) | 82.89 (0.31) | **92.84 (0.27)** | **79.69 (0.34)** |
| | EMD | 86.93 (0.28) | 44.03 (0.31) | **91.82 (0.14)** | 42.32 (0.31) | | - | - | - | - |
| | Sinkhorn | 86.93 (0.10) | 43.99 (0.37) | 91.61 (0.31) | **42.45 (0.09)** | | **94.42 (0.21)** | **83.51 (0.30)** | 91.84 (0.42) | 79.33 (0.38) |
| | SWD | 86.42 (0.26) | 43.35 (0.71) | 87.24 (0.29) | 42.39 (0.30) | | 94.41 (0.25) | 82.96 (0.17) | 83.58 (0.60) | 72.38 (0.43) |

For OT methods, we have $k$ class-conditioned clusters, resulting in $O(k^2)$ pairs of matrices as input for all OT methods. For each pair, the computation time of OT methods can range from linear to cubic with respect to $n$. For instance, each FastFT computation takes $O(nd)$ as shown in Theorem 3.2, so the total computation time becomes $O(k^2nd)$. In general, the computation time for all OT methods can be expressed as $O(k^2 \cdot \text{OT-time})$. Next, we elaborate on the "OT-time" term.

For EMD, the complexity is typically $O(n^3 \log n)$ in the past literature (Cuturi, 2013), which primarily focuses on the influence of the number of samples $n$. However, in our setting, because $d >> n$, we aim to emphasize the effect of $d$. In the time complexity expression for EMD, the $O(n^3 \log n)$ term arises from solving the linear programming problem, while the $O(nd)$ term accounts for computing the objective function $\langle \boldsymbol{P}, \boldsymbol{D} \rangle$. Due to the sparse nature of the linear programming solution, there are only $O(n)$ non-zero values in $\boldsymbol{P}$. Consequently, we compute $O(n)$ $d$-dimensional pairwise distances in $\boldsymbol{D}$, making the dot product computation require $O(nd)$ in total. Therefore, the overall complexity of EMD is $O(n(d + n^2 \log n))$.

To get the approximated flow matrix, Sinkhorn (Cuturi, 2013) mainly depends on the iterative matrix scaling algorithm. For each $I$ iteration, we normalize the row and column of a matrix of size $O(n^2)$, so the complexity for running this iterative algorithm is $O(n^2 I)$. Unlike in EMD, the approximated flow matrix $P$ derived from Sinkhorn is not sparse. Therefore, the dot product of the objective function will now take $O(n^2 d)$ in total. The total complexity of Sinkhorn is $O(n^2(d + I))$.

To obtain the approximated flow matrix, Sinkhorn (Cuturi, 2013) relies on the iterative matrix scaling algorithm. During each of the $I$ iterations, we normalize the rows and columns of a matrix of size $O(n^2)$, resulting in a complexity of $O(n^2 I)$ for the iterative algorithm. Unlike EMD, the approximated flow matrix $\boldsymbol{P}$ derived from Sinkhorn is not sparse. Consequently, computing the dot product for the objective function requires $O(n^2 d)$ in total. Therefore, the overall complexity of Sinkhorn is $O(n^2(d + I))$.

Table 18: Fine level generalization performance for each BREEDS setting.

| Dataset | Objective | sAcc | tAcc | sMAP |
|---------|-----------|------|------|------|
| L17 | Flat | 85.15 (0.30) | 46.31 (0.31) | 85.94 (0.18) |
|  | $\ell_2$ | 85.74 (0.69) | 47.50 (0.73) | 74.07 (0.61) |
|  | FastFT | 85.21 (0.38) | 46.47 (0.75) | 74.46 (0.16) |
|  | Sinkhorn | 85.80 (0.49) | **48.37 (0.69)** | 70.50 (0.53) |
|  | SWD | **85.88 (0.26)** | 47.50 (0.09) | **86.65 (0.51)** |
| N26 | Flat | 79.76 (0.27) | 36.06 (0.37) | 70.75 (1.14) |
|  | $\ell_2$ | 80.58 (0.54) | 37.08 (0.26) | 55.51 (0.50) |
|  | FastFT | 80.78 (0.52) | 36.69 (0.19) | 58.87 (0.42) |
|  | Sinkhorn | 81.74 (0.26) | **38.16 (0.14)** | 57.54 (0.92) |
|  | SWD | **81.79 (0.25)** | 37.71 (0.09) | **76.03 (0.05)** |
| E13 | Flat | 84.37 (0.24) | 56.04 (0.27) | 74.04 (0.34) |
|  | $\ell_2$ | 84.77 (0.18) | 55.69 (0.37) | 66.92 (0.16) |
|  | FastFT | **85.04 (0.15)** | 56.66 (0.44) | 67.96 (0.64) |
|  | Sinkhorn | 85.00 (0.15) | **57.14 (0.40)** | 64.47 (0.20) |
|  | SWD | 84.49 (0.18) | 56.82 (0.12) | **75.39 (0.54)** |
| E30 | Flat | 79.05 (0.23) | 42.35 (0.57) | 64.23 (0.17) |
|  | $\ell_2$ | **79.54 (0.21)** | 43.51 (0.23) | 54.92 (0.23) |
|  | FastFT | 79.29 (0.32) | 43.16 (0.42) | 54.49 (0.23) |
|  | Sinkhorn | 79.40 (0.33) | **43.82 (0.38)** | 53.66 (0.53) |
|  | SWD | 79.42 (0.52) | 43.28 (0.34) | **66.87 (0.45)** |

Table 19: Coarse level generalization performance for each BREEDS setting.

| Dataset | Objective | sAcc | tAcc | sMAP | tMAP |
|---------|-----------|------|------|------|------|
| L17 | $\ell_2$ | 93.13 (0.62) | 79.03 (0.29) | 93.93 (0.54) | 69.13 (0.42) |
|  | Flat | 92.68 (0.39) | 78.35 (0.37) | 76.66 (0.26) | 56.13 (0.36) |
|  | FastFT | **93.33 (0.26)** | 78.37 (0.47) | **92.11 (0.23)** | **65.44 (0.24)** |
|  | Sinkhorn | 93.11 (0.20) | **79.50 (0.43)** | 89.14 (0.36) | 63.88 (0.32) |
|  | SWD | 92.96 (0.41) | 78.66 (0.26) | 83.92 (0.38) | 59.66 (0.21) |
| N26 | $\ell_2$ | 88.94 (0.42) | 70.24 (0.50) | 89.59 (0.26) | 68.40 (0.52) |
|  | Flat | 88.44 (0.12) | 68.98 (0.41) | 61.20 (1.01) | 50.86 (0.16) |
|  | FastFT | 88.83 (0.75) | 69.43 (0.42) | **83.89 (0.40)** | 65.22 (0.75) |
|  | Sinkhorn | **89.94 (0.35)** | **70.70 (0.41)** | 83.14 (0.78) | **66.07 (0.85)** |
|  | SWD | 89.88 (0.34) | 69.93 (0.04) | 74.06 (0.57) | 58.50 (0.48) |
| E13 | $\ell_2$ | 97.80 (0.04) | 91.51 (0.01) | 98.25 (0.04) | 94.15 (0.09) |
|  | Flat | 97.53 (0.12) | 91.30 (0.16) | 83.89 (0.13) | 82.37 (0.31) |
|  | FastFT | **97.80 (0.01)** | 91.78 (0.13) | **97.69 (0.13)** | **93.21 (0.10)** |
|  | Sinkhorn | 97.63 (0.02) | **91.97 (0.14)** | 97.53 (0.23) | 92.77 (0.18) |
|  | SWD | 97.67 (0.13) | 91.69 (0.06) | 91.43 (0.72) | 87.71 (0.16) |
| E30 | $\ell_2$ | 97.07 (0.10) | 91.74 (0.10) | 98.24 (0.14) | 95.28 (0.24) |
|  | Flat | 96.75 (0.18) | 91.10 (0.15) | 76.12 (0.66) | 76.65 (0.74) |
|  | FastFT | 97.02 (0.07) | **91.98 (0.23)** | **97.65 (0.32)** | **94.90 (0.26)** |
|  | Sinkhorn | 96.99 (0.25) | 91.87 (0.22) | 97.54 (0.31) | 94.59 (0.18) |
|  | SWD | **97.12 (0.13)** | 91.55 (0.31) | 84.89 (0.74) | 83.66 (0.88) |

For FT (Backurs et al., 2020), the most computationally expensive step is the tree construction using the QuadTree algorithm. Backurs et al. (2020) demonstrate that this tree can be built in

$O(nd \log(d\Phi))$. Once the tree is constructed, as shown in Lemma 3.1 of Backurs et al. (2020), FT requires $O(n(d + \log(d\Phi)))$ to run the bottom-up algorithm and compute the objective function, which is less expensive than the tree construction. Thus, the total complexity of FT is $O(nd \log(d\Phi))$.

For Fast FlowTree, please refer to the proof of Thm. 3.2.

For TWD, we follow Algorithm 1 in Yamada et al. (2022). The first step is to construct a tree. To save computation, instead of using QuadTree, we leverage the same technique as in FastFT to use our augmented tree, making this step $O(1)$. For each of the $I$ iterations in supervised learning, we need to compute all entries in the pairwise distance matrix $D$, which requires $O(n^2 d)$. Thus, even when the tree structure is known, the total complexity of TWD is $O(n^2 dI)$.

Finally, we want to highlight that Fig.3 is evaluated with $k = 2$, where we vary the number of samples in one source and one target distribution. In this scenario, although the theoretical computational complexities for $\ell_2$ and Fast FlowTree are both $O(nd)$, we observe that $\ell_2$ scales better in practice. This is due to the effect of standard vectorization optimizations in common matrix operation packages, which leverage parallelism along the dimension of $n$.

### I.1 COMPARISON TO OTHER HIERARCHICAL BASELINES

In Fig.12, we compared OT-CPCC methods with other hierarchical objectives described in Sec.G, and we report the Fine/Coarse Metric Avg., which averages all source/target classification and retrieval metrics. Based on CPCC, fine, and coarse metrics, we observe that OT-CPCC methods behave similarly to $\ell_2$-CPCC, exhibiting higher CPCC values and stronger performance on coarse metrics, but lower fine metrics (though OT methods still outperform $\ell_2$ on fine metrics, as shown in Tab.5). In contrast, all other hierarchical baseline objectives perform more like flat cross-entropy methods, with the opposite trend—lower CPCC and coarse metrics but relatively higher fine metrics. Additionally, we excluded Rank loss from the correlation plot on the right side of Fig.12, as Tab.20 suggests that Rank loss is not suitable for any of the downstream tasks used in our evaluation, making it an outlier compared to the other objectives.

| Objective | CPCC |
|---|---|
| $\ell_2$ | 83.08 |
| FastFT | 93.85 |
| EMD | 89.60 |
| Sinkhorn | 93.81 |
| SWD | 73.17 |
| Flat | 22.13 |
| MTL | 37.07 |
| Curr | 22.70 |
| SumLoss | 27.48 |
| HXE | 22.65 |
| Soft | 50.90 |
| Quad | 24.01 |
| SHC | 30.68 |
| GC | 21.88 |
| Rank | 24.75 |

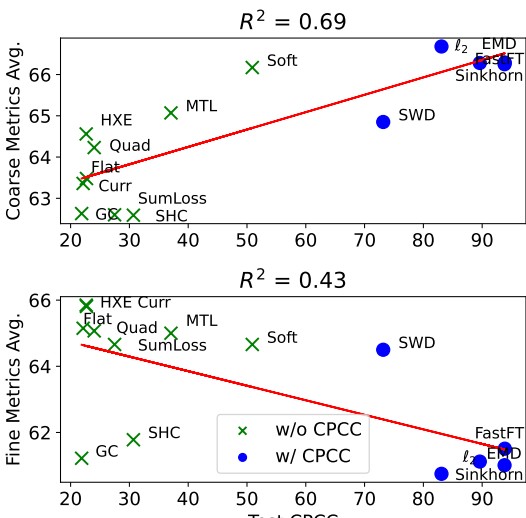

Figure 12: Left: Comparison of CPCC for hierarchical baselines on CIFAR100. Right: CPCC against coarse and fine metrics average. Rank is an outlier, so its value is removed from the visualization.

## J IMPLEMENTATION DETAILS FOR SYNTHETIC DATASETS EXPERIMENTS

**Computational Efficiency** We measured the computation time for different OT methods across datasets of varying sizes. This involved generating two random 128-dimensional datasets of identical shape and recording the wall-clock time for a single distance computation. This process was repeated

Table 20: Comparison of fine and coarse metrics for Flat vs. Rank Loss.

| Objective | Fine Metrics | | | Coarse Metrics | | | |
|---|---|---|---|---|---|---|---|
| | sAcc | tAcc | sMAP | sAcc | tAcc | sMAP | tMAP |
| Flat | 80.53 | 28.44 | 86.47 | 86.17 | 43.09 | 82.07 | 42.09 |
| Rank | 77.89 | 20.48 | 74.39 | 83.41 | 39.25 | 46.60 | 24.31 |

10 times, and the average time was reported. For illustrative purposes, we fixed the number of projections at 10 for SWD, and the regularization parameter at 10 for Sinkhorn, the same as in our experiment setting in Tab.4,5.

**Approximation Error** We evaluated the error of different OT approximation methods, along with the $\ell_2$ distance of the mean of two datasets, in approximating EMD. Two different distribution scenarios were considered:

- **Gaussian Distribution**: Here, both distributions were created as Gaussian distributions with different means (1 and 4, respectively) and the same standard deviation of 1.

- **Non-Gaussian Distribution**: For this scenario, we generated the dataset from Gaussian mixtures. One dataset combined Gaussian distributions with means of 0 and 5, while the other merged Gaussian distributions with means of 2 and 3, all of them with a standard deviation of 1.

Other experiment settings were consistent with those detailed in the Computational Efficiency paragraph above.

## K    COMPARISON OF RUNTIME OF OT-CPCC ON REAL-WORLD DATASETS

Table 21: Comparison of the wall clock runtime of OT-methods on real datasets for training one epoch. The results are averaged for three random seeds.

| Dataset | Objective | Per Epoch (s) | Dataset | Objective | Per Epoch (s) |
|---|---|---|---|---|---|
| | Flat | 85.88 (3.69) | | Flat | 147.67 (4.54) |
| | $\ell_2$ | 83.78 (9.80) | | $\ell_2$ | 149.39 (1.24) |
| *CIFAR10* | FastFT | 85.21 (4.26) | *INAT* | FastFT | 132.01 (3.24) |
| (b=128) | EMD | 87.70 (2.43) | (b=1024) | EMD | 234.68 (4.40) |
| | Sinkhorn | 85.65 (4.15) | | Sinkhorn | 165.69 (4.60) |
| | SWD | 92.90 (4.65) | | SWD | 152.52 (8.90) |

We include the wall clock run times for training different CPCC methods over one epoch in Tab.21, using two distinct settings: CIFAR10 with a batch size of 128 and model training from scratch, and INAT with a batch size of 1024 and model fine-tuning from an ImageNet checkpoint. Despite the differences in settings, we observe that FastFT consistently runs significantly faster than exact EMD computation and outperforms the other two approximation methods, SWD and Sinkhorn. This aligns with our theoretical analysis in Tab.1 and observations on synthetic data in Fig. 3. When comparing FastFT to $\ell_2$, we find that FastFT is not much slower, and in the case of INAT, it is even faster than $\ell_2$. Thus, we can confidently conclude that our proposed method, especially FastFT, is scalable.

Similar to $\ell_2$, one could argue for using another closed-form Wasserstein expression, the Bures-Wasserstein distance (as defined in Eq.7), in CPCC optimization. However, Bures depends on the covariance term of the source and target distributions, which requires at least $O(nd^2)$ computation. As shown in Tab.1, in $\ell_2$, calculating the mean of a class-conditioned cluster only requires $O(nd)$, and all other OT methods have a linear $d$ term in their time complexity. In CIFAR100, even with a large batch size of 1024, $n$ is approximately $1024/100 \approx 10$, assuming all classes are of equal size. Using ResNet18 as the backbone, $d = 512$, which is typically much larger than $n$. Therefore, the $d^2$ factor from computing the covariance matrix in Bures can make it prohibitively expensive in our case.

Table 22: Comparison of the wall clock runtime of Bures-CPCC vs. OT-CPCC on real datasets for training one batch. The results are averaged for three random seeds.

| Dataset | Objective | Per Batch (s) |
|---|---|---|
| CIFAR100 | $\ell_2$ | 0.35 (0.18) |
| | EMD | 1.94 (0.12) |
| | Bures | 33.77 (4.73) |

To verify this, we computed the running time for CIFAR100 (batch size = 128) for different methods in Tab.22, considering one batch update. The results are averaged over three random seeds.

As observed in Tab.22, Bures-Wasserstein is significantly slower than both $\ell_2$ and EMD due to its quadratic dependence on $d$. Therefore, for scalability concerns, covariance-based Wasserstein metrics are not well-suited to our problem setup.

