# OpenReview forum: "Learning Structured Representations by Embedding Class Hierarchy with Fast Optimal Transport"
_ICLR.cc/2025/Conference — ICLR 2025 Poster_

### Official Review · Reviewer_Tr4V · 2024-10-16

**Soundness:** 4
**Presentation:** 4
**Contribution:** 3
**Rating:** 8
**Confidence:** 4

**Summary:**

This paper tackles the problem of structured knowledge embedding in supervised learning, i.e., how to enforce that representations encode structures in the label space. An example of structure in label space is when labels are hierarchical (e.g., dog -> husky, bulldog, etc). Previous work has considered comparing class-conditional distributions through the Euclidean distance of the mean of their samples. However, as the authors remark, this choice is restrictive as it does not really capture all the information of a distribution (e.g., multi-modality, covariances, etc). The authors propose using the EMD to compare these class-conditionals, improving the learning of representations.

---

__Post-Rebuttal__

The authors did a good job on the rebuttal, and addressed all of my concerns. I also praise the initiative of running further experiments with the Bures-Wasserstein metric, which proved to be too computationally heavy for the task at hand (since $d \gg n$).

Overall, this is a strong submission, so I decided to increase my score from __6. Marginally above the acceptance threshold__ to __8. Accept, good paper__.

**Strengths:**

- The ideia is good. OT makes a good contribution to the considered problem, especially concerning how distributions are compared and embedded in the tree structure.

- The authors present a solid analysis of how to extend the $\ell_{2}$-CPCC framework to OT, especially concerning differentiating through linear programming.

- The authors' experiments show a solid improvement over previous framework under a variety of ways of computing Wasserstein distances.

**Weaknesses:**

- In Table 1 (Sec 2.2), I do not think that these complexities are straightforward to derive. Can the authors point to the reasoning to their derivation? For instance, EMD's complexity is tipically $\mathcal{O}(n^{3}\log n)$ instead of $\mathcal{O}(n(d+n^{2}\log n))$

- While I got what the authors are trying to express with prop. 3.1, and the result is true if stated correctly, I think the authors could rephrase it for better clarity. First, "$\ell_{2}$ for Gaussians" is not mathematically correct, since the $\ell_{2}$ norm or distance is defined for vectors. The closest term to this expression would be an $L_{2}$ norm or distance between probability distributions, but it is not equivalent to the $\ell_{2}$ distance between the means. I propose that the authors change the statement to "EMD reduces to $\ell_{2}$ distance between the means". Other ways of rephrasing this are possible, see my suggestions and questions below. Likewise, an appropriate reference to (Takatsu, 2011) would be interesting.

- In table 4, the objective "Flat" is not explained. From the overall context, I guess this corresponds to learning with no structure tree, but this is not clear to me.

**Questions:**

# Suggestions

- In Sec. 2.2, I suggest the authors to comment more specifically on the complexity of OT and its variants. I get that this is somewhat done in Table 1, but I did not find any references to this table. I think a clear discussion on the text would be beneficial (and, of course, a reference to the table).

- For better uniformity and clarity, the authors might want to change references to the EMD as a distance in favor of $2-$Wasserstein distance, which is what is actually used in the paper. For instance, in most tables the authors use EMD and terms involving the Wasserstein distance (TWD or SWD).

- In section 3.3, when authors are comparing the gradients of $\ell_{2}-$CPCC and OT-CPCC, I think the authors can make 2 further links, if they agree with the following discussion. First, the comparison of distributions' centroids can be understood as computing the MMD, with a linear kernel, between those distributions. This can further make the link with other probability metrics, and arguably, OT yields a richer geometry than this simplistic choice. Second, if one chooses to approximate OT through the Sinkhorn algorithm, letting the entropic regularization penalty $\epsilon \rightarrow +\infty$ leads to the trivial coupling $P_{ij} = 1/nm$. In this case, OT-CPCC boils down to the $\ell_{2}-$CPCC algorithm, modulo a scaling factor on the gradients. The authors can also cite (Feydy et al, 2019) which discusses these relations in depth.

# Questions

I have only one question, which I detail below.

Proposition 3.1 left me wondering if it would not be possible to run OT-CPCC with a parametric approximation of the classes through Gaussian measures. This way, instead of calculating EMD between classes, one would use the Bures-Wasserstein metric,

$\mathcal{W}_{2}(P,Q)^{2}  = \lVert \mu\_{P} - \mu\_{Q} \rVert\_{2}^{2}  +  Tr(\Sigma\_{P} + \Sigma\_{Q} - 2(\Sigma\_{P}^{1/2}\Sigma\_{Q}\Sigma\_{P}^{1/2})^{1/2}).$

With this approach, you could trade the complexity with respect the number of samples to a complexity with respect number of dimensions, which may be desirable depending on the problem at hand. Some authors have used this in other contexts (e.g., (Alvarez-Melis and Fusi, 2020)).

Furthermore, stressing the idea that class-conditionals are multi-modal, GMMs could be fit on these conditionals, then compute Mixture-Wasserstein (Delon and Desolneux, 2020) distances between the class-conditionals.

---

> ### Author Response · Authors · 2024-11-22
> **Reply to Reviewer Tr4V: Part 1**
>
> We sincerely thank the reviewer for acknowledging the merit of our idea, the contribution of OT in comparing and embedding distributions within the tree structure, the solid analysis of extending the ℓ2-CPCC framework to OT, and the strong experimental improvements demonstrated under various Wasserstein distance computations. Let us answer your questions and concerns below:
>
> > *In Table 1 (Sec 2.2), I do not think that these complexities are straightforward to derive. Can the authors point to the reasoning to their derivation? For instance, EMD's complexity is tipically $O(n^3\log n)$ instead of $O(n(d + n^2\log n))$.*
>
> Sorry for confusion. We fixed some minor mistakes in Table 1, and let us walk you through all time complexities listed in the updated Table 1.
>
> For simplicity, in the time complexity analysis, we assume each class conditioned cluster has the same size $n$, which implies batch size $b = nk$.
>
> For $\ell_2$, given one batch of batch size $b$, computing Euclidean centroids for all classes requires scanning through all $d$-dimensional data points once, which requires $O(bd) = O(nkd)$. We have at most $k$ classes in one batch, so computing the pairwise distances between class centroids will cost $O(k^2d)$. The total cost for $\ell_2$ is $O((nk+k^2)d)$.
>
> For OT methods, we have $k$ class conditioned clusters, so there are $O(k^2)$ pairs of matrices as input for all OT methods. For each pair, OT methods computation can take from linear time to cubic time w.r.t $n$. For example, each FastFT computation is $O(nd)$ as we shown in Theorem 3.2, so the total computation time is $O(k^2nd)$. In general, the computation time for all OT methods will be $O(k^2 \cdot \text{OT-time})$.
>
> Let us elaborate the “OT-time” part.
>
> For EMD, EMD’s complexity is typically $O(n^3\log n)$ in the past literature, mainly focusing on the influence of the number of samples $n$. However, in our setting, because $d >> n$, we want to emphasize the effect of $d$. In the expression of the time complexity for EMD, $O(n^3\log n)$ comes from solving the linear programming problem, and the $O(nd)$ term comes from the computation of the objective function $\left<\mathbf{P}, \mathbf{D}\right>$. Due to a sparse solution of linear programming, there are only $O(n)$ non-zero values in $\mathbf{P}$, we need to compute $O(n)$ $d$-dimensional pairwise distances in $D$, so the dot product takes $O(nd)$ in total. The total complexity of EMD is $O(n(d + n^2\log n))$.
>
> To get the approximated flow matrix, Sinkhorn mainly depends on the iterative matrix scaling algorithm. For each $I$ iteration, we normalize the row and column of a matrix of size $O(n^2)$, so the complexity for running this iterative algorithm is $O(n^2I)$. Unlike in EMD, the approximated flow matrix $P$ derived from Sinkhorn is not sparse. Therefore, the dot product of the objective function will now take $O(n^2d)$ in total. The total complexity of Sinkhorn is $O(n^2(d+I))$.
>
> For SWD, the first step is to map $d$-dimensional points to 1D by computing dot products of the data points with random vectors for $p$ repetitions, which involves $O(npd)$. For each of the $p$ projection, we need to sort the projected $n$ values, and compute the cumulative sum of the sorted values. Due to sorting, this step takes $O(np\log n)$. The total complexity of SWD is $O(np(d + \log n))$.
>
> For FT, the most expensive part is the tree construction using QuadTree algorithm. [1] shows that we can build this tree in $O(nd\log (d\Phi))$. After the tree is constructed, by lemma 3.1 in [1], FT requires $O(n(d + \log (d\Phi)))$ for running the bottom-up algorithm and computing the objective function, which is less expensive than the tree construction. The total complexity of FT is therefore $O(nd\log (d\Phi))$.
>
> For FastFT, please refer to the proof of Theorem 3.2.
>
> For TWD, we follow the Algorithm 1 in [2]. The first step is to construct a tree. Instead of using QuadTree, to save some computation, we can leverage the same technique in FastFT to use our augmented tree, so this step is $O(1)$. Now for each of $I$ iteration, for supervised learning, we need to compute all of the entries in the pairwise distance matrix $D$, which takes $O(n^2d)$. Therefore, even when tree structure is known, TWD will take $O(n^2dI)$.
>
> We included this detailed discussion of time complexities in App.I in the latest version.
>
> [1] Backurs, Arturs, et al. "Scalable nearest neighbor search for optimal transport." International Conference on machine learning. PMLR, 2020.
>
> [2] Yamada, Makoto, et al. "Approximating 1-wasserstein distance with trees." arXiv preprint arXiv:2206.12116 (2022).

---

> ### Author Response · Authors · 2024-11-22
> **Reply to Reviewer Tr4V: Part 2**
>
> > *While I got what the authors are trying to express with prop. 3.1, and the result is true if stated correctly, I think the authors could rephrase it for better clarity.*
>
> Thanks for pointing out this writing issue. We will take your suggestions and revise the statement as “EMD reduces to $\ell_2$ between means of Gaussian with the same covariance.” The change is reflected in line 181-183 of the latest version.
>
> > *In table 4, the objective "Flat" is not explained. From the overall context, I guess this corresponds to learning with no structure tree, but this is not clear to me.*
>
> Yes, “Flat” corresponds to standard cross entropy learning without using any hierarchical information. We now added one extra line for clarification in line 332.
>
> > *For better uniformity and clarity, the authors might want to change references to the EMD as a distance in favor of 2−Wasserstein distance, which is what is actually used in the paper. For instance, in most tables the authors use EMD and terms involving the Wasserstein distance (TWD or SWD).*
>
> Thanks for your suggestion! We understand your concern about consistency of naming. As we mentioned in App.B in the proof of Prop.3.1, EMD is equivalent to 1-Wasserstein distance, using $\ell_2$ as the ground metric, for discrete measures. The reason why we use the name of EMD is because in our paper, we restrict our discussion only on discrete measures where the distance metric can be derived from a linear programming problem in Eq.3, while the name of 1-Wasserstein distance can be applied to a more general case where $\mu$ and $\nu$ can be continuous. With the linear programming setup in Eq.3, we can explain gradient analysis in Sec 3.3 easier.
>
> > *In section 3.3, when authors are comparing the gradients of ℓ2−CPCC and OT-CPCC, I think the authors can make 2 further links, if they agree with the following discussion.*
>
> We appreciate your insightful comments! We added the discussion of the relationship between MMD and $\ell_2$ in line 43-44.
>
> However, the equivalence between Sinkhorn with $\epsilon \to \infty$ and $\ell_2$ does not hold in general. Assume we have two matrices $X \in \mathbb{R}^{n \times d}$ and $Y \in \mathbb{R}^{m \times d}$ representing two class conditioned features. When $\epsilon \to \infty$, Sinkhorn’s flow matrix are uniformly distributed with each entry to be $1/mn$, so the dot product of flow matrix $\mathbf{P}$ and pairwise distance matrix $\mathbf{D}$ is the average of all sample pairwise distances $ \frac{1}{nm} \sum_{i,j} \mathbf{D}(X_i, Y_j)$. On the other hand, $\ell_2$ is the distance between two class means, which is $\|\frac{1}{n}\sum_{i}X_i - \frac{1}{m}\sum_{j}Y_j\|$. These two expressions are only equivalent when all points in each matrix are equidistant from each other, which does not hold in general.

---

> ### Author Response · Authors · 2024-11-22
> **Reply to Reviewer Tr4V: Part 3**
>
> > *Run OT-CPCC with Bures-Wasserstein or Mixture-Wasserstein*
>
> Yes, this is a natural extension of the $\ell_2$ CPCC method. We first restate the expression of both metrics. Bures-Wasserstein is defined as
> $$\mathcal{W}_{2}(P,Q)^{2}  = \|\mu_P - \mu_Q\|_2^2 + \text{Tr}(\Sigma_P + \Sigma_Q - 2(\Sigma_P^{1/2}\Sigma_Q\Sigma_P^{1/2})^{1/2}),$$
>
> and the objective function of Mixture-Wasserstein [1] is
>
> $$\min \sum w_{PQ} \mathcal{W}_{2}(P,Q)^{2}.$$
>
> We can observe that both metrics depend on the covariance term which requires at least $O(nd^2)$ computation, and Mixture-Wasserstein requires additional time for fitting GMM. Taking the mean of a class conditioned cluster requires $O(nd)$. All other OT methods only have a linear $d$ term in the big-O term too. In practice, $n$ is the size of a class conditioned cluster. In CIFAR100, if batch size = 1024, $n$ is approximately $1024/100 \approx 10$  if all classes have the same size. If we use ResNet18 as the backbone, $d = 512$ which is typically much larger than $n$. Therefore, the factor of $d$ from computing the covariance matrix can be very expensive in our case. To verify this statement, we computed the running time for CIFAR100 (batch size = 128) for different methods below, for one batch update. The results are averaged over three random seeds.
>
> | Dataset   | Objective | Per Batch (s) |
> |-----------|-----------|---------------|
> |           | $\ell_2$        | 0.35 (0.18)   |
> | **CIFAR100**  | EMD       | 1.94 (0.12)   |
> |           | Bures     | 33.77 (4.73)  |
>
> We can observe that Bures-Wasserstein is much slower than both $\ell_2$ and EMD. Therefore, for scalability concern, all covariance based metrics do not fit our problem setup very well.
>
> [1] Delon, Julie, and Agnes Desolneux. "A Wasserstein-type distance in the space of Gaussian mixture models." SIAM Journal on Imaging Sciences 13.2 (2020): 936-970.

---

> > ### Comment · Reviewer_Tr4V · 2024-11-25
> > **Response to authors' rebuttal**
> >
> > Dear authors,
> >
> > Thank you for the clarifications. Here are some considerations on your rebuttal,
> >
> > __On complexities.__ With the rebuttal, the complexities in Table 1 are much more clearer. Thank you.
> >
> > __On connections with other probability metrics.__ Indeed, your assessment is correct about the connection between the MMD and entropic OT. Thank you for considering my comment about the MMD and $\ell_{2}$.
> >
> > __On using Bures-Wasserstein.__ I thank the authors for taking the initiative of performing extra experiments with my suggestion. If I may ask, I would also like to ask the authors to report the performance of Bures-Wasserstein in comparison with the EMD. Other than this, I am satisfied with the authors considerations.
> >
> > ----
> >
> > In general, I consider that my concerns were properly addressed and I will increase my score accordingly.

---

> ### Author Response · Authors · 2024-11-25
>
> Thanks for carefully reading our rebuttal and we appreciate your insightful feedback. We just updated the submission and included the discussion about the possibility of using Bures-Wasserstein-CPCC in our rebuttal into App.K (line 1613-1634). Because of the scalability issue for Bures-Wasserstein, it is unfortunately difficult to train with full epochs to evaluate other generalization/distortion metrics specifically for Bures-Wasserstein. Feel free to let us know if you have any remaining suggestions, questions, or concerns. We would be happy to continue the discussion.

---

> > ### Comment · Reviewer_Tr4V · 2024-11-26
> > **Response to Official Comment**
> >
> > I understand. Thank you for the experiments anyways, and congratulations on your work.

---

### Official Review · Reviewer_BGq8 · 2024-11-01

**Soundness:** 3
**Presentation:** 3
**Contribution:** 2
**Rating:** 6
**Confidence:** 3

**Summary:**

The paper addresses the challenge of embedding hierarchical structures in feature representations for supervised classification tasks. It builds upon existing work that uses the Cophenetic Correlation Coefficient (CPCC) to enforce hierarchical structure alignment but innovates by employing the Earth Mover’s Distance (EMD) within this framework. The authors also propose a new, efficient algorithm, Fast FlowTree (FastFT), to approximate EMD in linear time, making it feasible to apply OT-based regularization to large datasets.

**Strengths:**

1. The paper is well-written and easy to follow.
2. The paper makes a theoretical extension by generalizing CPCC with EMD, allowing the model to handle more complex, non-unimodal distributions that Euclidean-based methods struggle with. This extension is theoretically sound and shows how OT can capture nuanced hierarchical structures in data.

**Weaknesses:**

1. The improvement of FastFT and EMD, which appears to be the paper’s key contribution, is modest on empirical datasets, showing less than a 1% gain on average across three random seeds.
2. The computation time for empirical datasets is not reported.
3. The paper is a follow-up to the CPCC work by Zeng et al. (2022), with the addition of EMD. The novelty of introducing Fast FlowTree is somewhat limited.

**Questions:**

While the paper proposes FastFT for computational efficiency, can the authors provide a clear comparison of computation time on empirical datasets?

---

> ### Author Response · Authors · 2024-11-22
> **Reply to Reviewer BGq8: Part 1**
>
> We sincerely thank the reviewer for appreciating the clarity of our writing and recognizing the theoretical extension of CPCC with EMD, which enables handling complex, non-unimodal distributions and effectively captures nuanced hierarchical structures in data. Let us answer your questions and concerns below:
>
> > *The improvement of FastFT and EMD, which appears to be the paper’s key contribution, is modest on empirical datasets, showing less than a 1% gain on average across three random seeds.*
>
> We understand your concern. We want to clarify that the 1% gain is only for the accuracy metrics in Table 4 and 5. However, on retrieval metrics in Table 4 and 5 (MAP), we can observe that the gain is significant, and typically much larger than 1%. Besides, when the multi-mode distribution assumption is valid (including CIFAR10, CIFAR100, INAT, based on GMM test in Table 5), we can see a huge improvement in the preciseness of hierarchical information in Table 3.
>
> > *The computation time for empirical datasets is not reported.*
>
> We include the wall clock run time of different CPCC methods training for one epoch. The result is averaged for three seeds, and we use two very different settings, CIFAR10 with batch size = 128 and model pretraining from scratch, and INAT with batch size = 1024 and model fine-tuned from ImageNet1K checkpoint. The table below summarizes the time comparison:
>
> | Dataset  | Objective | Per Epoch (s)  | Dataset  | Objective | Per Epoch (s)  |
> |----------|-----------|----------------|----------|-----------|----------------|
> |          | Flat      | 85.88 (3.69)   |          | Flat      | 147.67 (4.54)  |
> |          | $\ell_2$        | 83.78 (9.80)   |          | $\ell_2$        | 149.39 (1.24)  |
> | **CIFAR10**  | FastFT    | 85.21 (4.26)   | **INAT**     | FastFT    | 132.01 (3.24)  |
> | *(b=128)*  | EMD       | 87.70 (2.43)   | *(b=1024)* | EMD       | 234.68 (4.40)  |
> |          | Sinkhorn  | 85.65 (4.15)   |          | Sinkhorn  | 165.69 (4.60)  |
> |          | SWD       | 92.90 (4.65)   |          | SWD       | 152.52 (8.90)  |
>
> Although two datasets have very different settings, we can observe that FastFT consistently runs much faster than exact EMD computation, better than other two approximation methods SWD and Sinkhorn, which matches our theoretical analysis in Table 1. For the comparison between FastFT and $\ell_2$, FastFT isn’t much slower than $\ell_2$, and in INAT, FastFT is even faster than $\ell_2$. We can therefore safely conclude that our proposed method, especially FastFT, is scalable.
>
> We include this table in App.K of the paper.
>
> > *The paper is a follow-up to the CPCC work by Zeng et al. (2022), with the addition of EMD. The novelty of introducing Fast FlowTree is somewhat limited.*
>
> We want to first emphasize that from our motivating example (Fig. 1), OT-CPCC is well-inspired by very relevant practical multi-mode scenarios and generalizes the previous $\ell_2$-CPCC algorithm as a special case, thereby offering novel, stronger and more flexible solutions overall.
>
> In addition to proposing the novel OT-CPCC method for learning structured representation, we also
> * Proposed the novel FastFT algorithm which is a very high-quality approximation of exact EMD distance. To our best knowledge, we do not aware of the FastFT algorithm being proposed elsewhere under the context of EMD approximation.
> * Discovered the surprisingly close connection between FastFT, FlowTree, and the 1d-OT transportation algorithm.
> * Added a novel comprehensive forward/backward theoretical analysis of $\ell_2$/OT-CPCC to prove our method is a generalization of previous $\ell_2$-CPCC.
>
> We therefore believe we have made several original contributions that span both the research areas of structured representation learning and optimal transport. We expect our novel FastFT algorithm to have a much broader impact in many scenarios that require Wasserstein computation.

---

> > ### Comment · Reviewer_BGq8 · 2024-11-30
> >
> > Thank you to the authors for addressing my concerns. I am satisfied with the clarifications provided and have updated my score accordingly.

---

### Official Review · Reviewer_LYtZ · 2024-11-04

**Soundness:** 3
**Presentation:** 3
**Contribution:** 3
**Rating:** 8
**Confidence:** 2

**Summary:**

This paper presents a method for learning structured representations by embedding class hierarchies using Optimal Transport (OT) in the Cophenetic Correlation Coefficient (CPCC) framework. Traditional methods use Euclidean distances between class means to represent hierarchical information, which can be inaccurate for multi-modal distributions. The authors propose using Earth Mover’s Distance (EMD) to address this limitation and introduce Fast FlowTree (FastFT), a computationally efficient OT approximation algorithm for CPCC that operates in linear time.

**Strengths:**

The proposed algorithms are novel and scalable. The proposed method leads to performance improvements across multiple datasets. The work is impactful for applications needing hierarchical class embedding.

**Weaknesses:**

1. I feel section 3.3 can be simplified by moving some derivations and technical discussion to the appendix.
2. Label embedding methods have been employed in extreme classification (see datasets: http://manikvarma.org/downloads/XC/XMLRepository.html). The authors may want to experiment on those datasets to verify scalability and compare to existing embedding methods.

**Questions:**

see Weaknesses

---

> ### Author Response · Authors · 2024-11-22
> **Reply to Reviewer LYtZ: Part 1**
>
> We sincerely thank the reviewer for highlighting the novelty and scalability of our algorithms, the performance improvements across multiple datasets, and the impact of our work on applications requiring hierarchical class embedding. Let us answer your questions and concerns below:
>
> > *I feel section 3.3 can be simplified by moving some derivations and technical discussion to the appendix.*
>
> Thank you for your suggestion to simplify Section 3.3 by moving some content to the appendix. We feel the current level of detail is important for maintaining the paper’s flow and clarity. However, we will revisit this section if space adjustments are needed to address other reviewers’ feedback.
>
>
> > *Label embedding methods have been employed in extreme classification (see datasets: http://manikvarma.org/downloads/XC/XMLRepository.html). The authors may want to experiment on those datasets to verify scalability and compare to existing embedding methods.*
>
> Thanks for suggesting these datasets! Extreme classification sounds like a very interesting direction to apply our method. However, we want to show that it is nontrivial to apply CPCC in the extreme classification setting:
>
> Extreme classification deals with multi-label classification problems involving an exceptionally large number of labels. Instead of a multi-label classification setting, our hierarchical classification in Sec.4.3 is a multi-class setting where each data point only has one fine or one coarse label for fine and coarse level evaluations. To apply CPCC for multi-label classification, since each data point can have multiple labels, for a pair of data point, the “tree distance” between two data points cannot be the shortest distance between only *two* class nodes on the label hierarchy anymore. Some modifications of CPCC must be made, and we would like to leave it as future work.
>
> Besides, we want to show the difference between label embedding methods and OT-CPCC methods. For label embedding methods, each label is mapped to some high dimensional vector for downstream tasks: for example in the standard multi-class classification, we typically use one-hot vector embeddings. While $\ell_2$-CPCC falls into this category, OT-CPCC methods are not label embedding methods. Instead of representing one class with a Euclidean centroid, OT-CPCC methods intentionally avoided this step by considering a more nuanced relationship between labels. OT-CPCC depends on pairwise relationships between data points from two classes, to include a more complicated distributional geometry during learning. Since we rely on data point relations, there is no one-to-one correspondence between a label and a representation.
>
> We include the discussion about the difference between label embedding methods/learning label representation vs. structured representation learning in App.B with related works.

---

### Official Review · Reviewer_rpnu · 2024-11-04

**Soundness:** 3
**Presentation:** 3
**Contribution:** 3
**Rating:** 6
**Confidence:** 3

**Summary:**

The paper extends the work (Zeng et al. 2022) for learning structured representation by introducing the Earth Mover's Distance (EMD) to measure the pairwise distances among classes in embeddings. Moreover, a so-called Fast FlowTree algorithm is proposed to improve the computation efficiency, which is linear in the size of the dataset. Experiments show promising results.

**Strengths:**

+ It extends the $\ell_2$ norm based Cophenetic Correlation Coefficient (CPCC) (Zeng et al. 2022) to OT-CPCC by introducing EMD, which is essentially equivalent to assign different importance to each pair of data points when computing the gradients.
+ It presents a linear time approximation algorithm to solve the OP problem for learning the hierachical representation.
+ Extensive experiments are provided.

**Weaknesses:**

1.  To replace the $\ell_2$ distance between two class-centroids, the EMD is introduced based on representing class label by all samples in the class. If the number of samples in each class is large, then solving a single EMD is expensive in time cost. Notice that the distances are computed for pairwise classes, thus the times to compute EMD is quadratic to the number of classes. If the number of classes is large, then it is extremely expensive to compute the EMD based hierarchy information among embeddings.  While different approximation methods can be used, it is not clear about the computation cost, compared to the baseline methods.

2. In Table 3, the preciseness of hierachical information listed show that in most cases, the presented FastFT based method cannot beat the $\ell_2$ norm based method. How to interprete the results in Table 3?

**Questions:**

1. While different approximation methods can be used, it is not clear about the computation cost, compared to the baseline methods.

2. In Table 3, the preciseness of hierachical information listed show that in most cases, the presented FastFT based method cannot beat the $\ell_2$ norm based method. How to interprete the results in Table 3?

3. Using AIC to estimate the number of modality of the classes sounds interesting.  Is there any more intuitive evidence to support the existence of multi-mode in feature distribution? Or,  whether the identified multi-mode classes could be visualized by e.g., t-SNE or UMAP?

---

> ### Author Response · Authors · 2024-11-22
> **Reply to Reviewer rpnu: Part 1**
>
> We sincerely thank the reviewer for appreciating our extension of CPCC to OT-CPCC using EMD, the proposed linear-time algorithm for solving the OP problem, and the extensive experiments validating our approach. Let us answer your questions and concerns below:
>
> > *To replace the $\ell_2$ distance between two class-centroids, the EMD is introduced based on representing class label by all samples in the class. If the number of samples in each class is large, then solving a single EMD is expensive in time cost. Notice that the distances are computed for pairwise classes, thus the times to compute EMD is quadratic to the number of classes. If the number of classes is large, then it is extremely expensive to compute the EMD based hierarchy information among embeddings. While different approximation methods can be used, it is not clear about the computation cost, compared to the baseline methods.*
>
> Yes indeed, all EMD and all its approximation OT methods will be more expensive than the $\ell_2$ method. We made some minor mistakes in computing the time complexity of $\ell_2$ in the first submission. Now we updated the corrected results in Tab.1. Let us elaborate:
>
> For simplicity, in the time complexity analysis, we assume each class conditioned cluster has the same size $n$, which implies batch size $b = nk$.
>
> For $\ell_2$, given one batch of batch size $b$, computing Euclidean centroids for all classes requires scanning through all $d$-dimensional data points once, which requires $O(bd) = O(nkd)$. We have at most $k$ classes in one batch, so computing the pairwise distances between class centroids will cost $O(k^2d)$. The total cost for $\ell_2$ is $O((nk+k^2)d)$.
>
> For OT methods, we have $k$ class conditioned clusters, so there are $O(k^2)$ pairs of matrices as input for all OT methods. For each pair, OT methods computation can take from linear time to cubic time w.r.t $n$. For example, each FastFT computation is $O(nd)$ as we shown in Theorem 3.2, so the total computation time is $O(k^2nd)$. In general, the computation time for all OT methods will be $O(k^2 \cdot \text{OT-time})$.
>
> Since all OT methods are at least $O(k^2nd)$, and $O(k^2nd) \geq O((nk+k^2)d)$, it can be expensive to use exact EMD, and thus we propose using approximation methods to close this gap.
>
> In practice, however, for the standard batch size like 128/1024, in Fig.3 left, we can see when $k$ is fixed ($k$ = 2 in Fig.3), the difference between our proposed method, FastFT, and $\ell_2$ is much smaller than the gap between $\ell_2$ and other methods. Note that in Fig. 3, $\ell_2$ looks like a constant time w.r.t. $n$. This might be because the mean operation in Pytorch is well-vectorized, so the effect of $n$ is improved due to parallelism. Besides, we include the wall clock time of all CPCC methods for running one epoch below:
>
> | Dataset  | Objective | Per Epoch (s)  | Dataset  | Objective | Per Epoch (s)  |
> |----------|-----------|----------------|----------|-----------|----------------|
> |          | Flat      | 85.88 (3.69)   |          | Flat      | 147.67 (4.54)  |
> |          | $\ell_2$        | 83.78 (9.80)   |          | $\ell_2$        | 149.39 (1.24)  |
> | **CIFAR10**  | FastFT    | 85.21 (4.26)   | **INAT**     | FastFT    | 132.01 (3.24)  |
> | *(b=128)*  | EMD       | 87.70 (2.43)   | *(b=1024)* | EMD       | 234.68 (4.40)  |
> |          | Sinkhorn  | 85.65 (4.15)   |          | Sinkhorn  | 165.69 (4.60)  |
> |          | SWD       | 92.90 (4.65)   |          | SWD       | 152.52 (8.90)  |
>
> We can observe that: first, the gap between FastFT and $\ell_2$ isn’t too large, due to the different settings of batch size ($b$) , number of classes per batch ($k$), and the size of each class-conditioned cluster ($n$); second, we can also observe the difference between OT approximation methods approximately matches our time complexity analysis in Tab.1 and Fig.3 left.
>
> We included the details of time complexity analysis in App.I, and this empirical run time table in App.K.

---

> > ### Comment · Reviewer_rpnu · 2024-11-25
> >
> > The reviewer is satisifed to read the clarification on the computation complexity and the breaddown description for the compuation time cost.

---

> ### Author Response · Authors · 2024-11-22
> **Reply to Reviewer rpnu: Part 2**
>
> > *In Table 3, the preciseness of hierachical information listed show that in most cases, the presented FastFT based method cannot beat the ℓ2 norm based method. How to interprete the results in Table 3?*
>
> The preciseness of hierarchical information depends on the multi-modality of the class conditioned features. For all datasets, when we see a “FastFT CPCC lower than $\ell_2$ CPCC,” it only happens for BREEDS settings. From Figure 5 right, the “multimodality-ness” of 4 BREEDS dataset is 1.25, which is almost close to Gaussian. When the features are Gaussian, from Proposition 3.1 we know that $\ell_2$ and EMD are equivalent if all class conditioned features share the same covariance. The minor performance drop can be either explained by the approximation error of FastFT (w.r.t EMD), or by the violation of the shared covariance assumption. For other datasets with high modality, we can observe FastFT embeds the tree information very well.
>
> We included this explanation in line 487-489 of the paper.
>
> > *Using AIC to estimate the number of modality of the classes sounds interesting. Is there any more intuitive evidence to support the existence of multi-mode in feature distribution? Or, whether the identified multi-mode classes could be visualized by e.g., t-SNE or UMAP?*
>
> Thanks for suggesting this! We included UMAP in Fig.6, yet only for visualizing the difference between different learned representations qualitatively. We thought about using t-SNE/UMAP for multi-modality measurement, but we have several concerns, such as:
> * t-SNE/UMAP are not quantitative, because it is difficult to numerically determine the exact number of modes directly from visualization. Our current GMM based method, on the other hand, tells us the approximate number of modes.
> * t-SNE/UMAP are sensitive to parameters.
> * Since t-SNE/UMAP can be only visualized in 2D/3D, we may create some noisy artifacts in the process of dimensionality reduction. We directly learn GMM on high-dimensional data.
>
> Therefore, we still think our current AIC based method might be a better choice, and we can use the result to get some insights to explain some phenomenon, such as the performance drop of preciseness of hierarchical information as you mentioned above.

---

> > ### Comment · Reviewer_rpnu · 2024-11-25
> >
> > The reviewer is satisifed with the interpretation to the results in Table 3. But, the reviewer still has some concerns on the way to use AIC to estimate the number of modality of the classes. To be more specific, using GMM to high dimensional data is somehow questionable. When fitting a GMM to the distribution of the data, whatever the distribution of the data looks like, the learned components could alway be returned. However, whether the learned modality relates to the natural structure of the data of a specific class, is might not relevant. The reviewer checked Fig. 6, but cannot find any clue.  Since that the rating is already positve, the reviewer would like to keep it, but still be happy to see more reliable way to check the number of the modality of the classes.

---

> > > ### Author Response · Authors · 2024-11-25
> > >
> > > Thanks for carefully reading our review. Regarding the multi-modality test, although the training likelihood of GMM will always increase as $k$ grows, since AIC penalizes the number of parameters ($k$, in our case), we selected the optimal $k$ based on AIC that avoids any artifacts due to overfitting. Besides, the lowest AIC point often coincides with the elbow point on the curve that plots AIC against the number of components, which aligns with this standard model selection criteria. For Fig. 6, we mainly use it to compare the coarse level pattern between $\ell_2$ and EMD to support observations in Tab.4 and Tab.5. And due to the limitations we mentioned in the rebuttal of TSNE/UMAP, we do not use it to verify the multi-modal assumption.
> > >
> > > It might be difficult to determine the ground truth of the “natural structure of the data of a specific class” when the data is high-dimensional. If you have any alternative proposals for the multi-modality test, we would be happy to continue this discussion.

---

### Official Review · Reviewer_1XYv · 2024-11-08

**Soundness:** 3
**Presentation:** 3
**Contribution:** 2
**Rating:** 5
**Confidence:** 3

**Summary:**

The paper aims to improve classification by incorporating similarities among classes. These semantic relationships between labels are introduced into the original problem as a regularizer. The paper assumes a hierarchical structure in the labels. Building on the work of Zeng et al. (2022), the authors replace Euclidean distances between class means with optimal transport (OT) distances between label distributions. To reduce the computational burden of OT distance, they use a modified version of a tree-based method called FlowTree, which relies on a predefined tree structure. The paper also introduces a differentiable method for training with their tree-based OT distance. Experimental results for the classification problem using the OT-based regularization method show improvements compared to the method in Zeng et al. (2022). However, the proposed FastFT regularization falls behind the performance of some other OT-based methods.

**Strengths:**

- The paper employs optimal transport (OT) distance to measure the distributions of labels rather than relying solely on the distance between class means. By capturing distributional differences, this approach aims to provide a more nuanced measure of class similarity.

- To address the high computational complexity typically associated with OT distance, the authors implement a tree-based approximation method. This approach enables more efficient computation while maintaining the benefits of OT-based similarity measures.

- Experimental results demonstrate that accounting for the distributional nature of labels enhances performance in tasks such as classification and class representation learning. This distribution-based approach proves beneficial in capturing class relationships more effectively than mean-based methods.

**Weaknesses:**

- Theorem 3.2 appears to be a key contribution, as it establishes that Algorithms 1 and 2 can approximate the Earth Mover's Distance (EMD). However, the proof is vague and leaves several critical points open to interpretation. I found it challenging to follow the complete argument leading to the stated approximation result. Including a detailed proof sketch could greatly enhance clarity and strengthen the manuscript.

- Similarly, while Lemma 3.4 provides the derivative of the EMD, the result itself is relatively straightforward. More importantly, it is unclear whether the same derivative result directly applies to FastTree. A formal proof confirming that this result holds for FastTree would significantly enhance the paper's contribution and rigor.

- Another issue arises with the reliance on an assumed ground-truth tree structure. In most cases, this ideal tree is unknown or impractical to obtain, even if the underlying assumptions hold. As a result, multiple approximated trees may be used instead. The paper does not adequately address how gradient computations would handle these varied approximations, which is an important practical consideration.

- Finally, a literature review on learning label representations, accompanied by illustrative examples, would provide valuable context and motivation for the study. This would help readers appreciate the importance of this approach within the broader scope of label representation learning.

**Questions:**

See  the Weakness section.

---

> ### Author Response · Authors · 2024-11-22
> **Reply to Reviewer 1XYv: Part 1**
>
> We sincerely thank the reviewer for recognizing our use of OT distance to capture distributional differences, the efficient tree-based approximation method addressing OT’s computational complexity, and the experimental validation showing improved performance in classification and class representation learning. Let us answer your questions and concerns below:
>
> > *Theorem 3.2 appears to be a key contribution, as it establishes that Algorithms 1 and 2 can approximate the Earth Mover's Distance (EMD). However, the proof is vague and leaves several critical points open to interpretation. I found it challenging to follow the complete argument leading to the stated approximation result. Including a detailed proof sketch could greatly enhance clarity and strengthen the manuscript.*
>
> Sorry for confusion. Let us clarify that the purpose of Theorem 3.2 is to show the equivalence between Alg.1 and Alg.2 under the construction of the augmented tree (Fig. 2), rather than showing any of these algorithms can approximate the Earth Mover’s Distance.
> The key idea is as follows: we can observe that the blue code blocks, the update rule of the flow matrix, between Alg.1 and Alg.2 are almost identical. By [1], Alg.2 provides the optimal flow matrix for the optimal transport problem using the tree metric as the ground metric in $\mathbf{D}$. From Fig.2, the tree distance between two samples in the augmented tree are always the same, so the order of seeing any vertex within source or target distribution does not matter. This reduces a FlowTree problem in Alg.2 to a flat 1d OT problem in Alg.1, where the major difference with Alg.2 is we do not need to pass the flow from the leaf to the root node in Alg.1.
>
> Due to the limitation of space, we included a more detailed proof sketch in line 854-861 in the current version, and made a clarification about the statement of this theorem in line 210-214.
>
> Additionally, the approximation error of FT and FastFT is an open question, and we want to leave it for further theoretical analysis in the future. In FlowTree [2], the authors analyze the approximation factor of FastFT only in the nearest neighbor setup, where they only rely on the fact that $\text{EMD} = \left<\mathbf{P}^*_{\ell_2}, \mathbf{D}_{\ell_2}\right> \leq \left<\mathbf{P}_{\text{FT}}, \mathbf{D}_{\ell_2}\right> = \text{FT}$ by the definition of the linear programming expression of EMD. However, the exact approximation factor/error or FT is unknown.
>
> [1] Kalantari, Bahman, and Iraj Kalantari. "A linear-time algorithm for minimum cost flow on undirected one-trees." Combinatorics Advances. Boston, MA: Springer US, 1995. 217-223.
>
> [2] Backurs, Arturs, et al. "Scalable nearest neighbor search for optimal transport." International Conference on machine learning. PMLR, 2020.
>
> > *Similarly, while Lemma 3.4 provides the derivative of the EMD, the result itself is relatively straightforward. More importantly, it is unclear whether the same derivative result directly applies to FastTree. A formal proof confirming that this result holds for FastTree would significantly enhance the paper's contribution and rigor.*
>
> Sorry for being vague about the differentiability of FastFT. For FastFT, the flow matrix $\mathbf{P}$ does not depend on the model parameters $\theta$. Instead, from Alg.1 and Alg.2, we can observe that it only depends on the structure of the input (augmented) label tree. Therefore, in the computation of the objective function $\left<\mathbf{P}, \mathbf{D}(\theta)\right>$, $\mathbf{P}$ is treated as a constant with respect to the model parameters $\theta$, and the backward gradient flow does not pass through $\mathbf{P}$. So, we can still use $\mathbf{P}\cdot\frac{\partial\mathbf{D}}{\partial \theta}$ for the back propagation for Fast FlowTree. This expression has a format similar to EMD’s partial derivative.
>
> We include this explanation in line 295-297 of the current version.

---

> ### Author Response · Authors · 2024-11-22
> **Reply to Reviewer 1XYv: Part 2**
>
> > *Another issue arises with the reliance on an assumed ground-truth tree structure. In most cases, this ideal tree is unknown or impractical to obtain, even if the underlying assumptions hold. As a result, multiple approximated trees may be used instead. The paper does not adequately address how gradient computations would handle these varied approximations, which is an important practical consideration.*
>
> We understand your concern about using ideal trees. We want to emphasize that our method only relies on the existence of some tree, not necessarily ideal.
> * First, there is no way to verify the tree given in the dataset is ideal. For example, in CIFAR100, although we know fish and mammals are two different species biologically, we cannot notice a huge structural difference visually on fish and dolphins where the latter is a type of mammal. The given CIFAR100 hierarchy relies on some factual knowledge about nature, but it is hard to verify if this is ideal for the structured representation learning problem.
> * We can use some approximated versions of the trees in OT-CPCC. We included the Effect of Tree Depth in App.H.1 and Effect of Tree Weight in App.H.2.
>     * In the setting of Effect of Tree Depth, we started from tree depth = 3, which is the ground truth root-coarse-fine hierarchy. We create synthetic mid/coarser levels for tree depth = 4,5, and this modification can be seen as an approximation of the original depth = 3 ground truth tree. From Fig.10, we can observe that OT-CPCC successfully embedded the approximated, or synthetic information well. We also attached the several performance metrics in Tab.13 and at the very end of this reply. We can observe that all CPCC values are high, which matches our visualized pattern in Fig.10. For the fine and coarse accuracy, we didn’t observe a huge disadvantage of using approximated trees.
>     * In the setting of Effect of Tree Weights, we started from weight x1, which is the ground truth hierarchy. To create approximated hierarchical trees, we adjusted the distance within the fine classes under the coarse label of aquatic mammals by multiplying the original weights by x2, and x4. We include the visualization in Fig.11, and performance metrics in Tab.14. The conclusions are similar to the experimental results for different tree depths.
> * Since all trees are treated as a constant in our learning pipeline, i.e, in CPCC computation, only the feature distances are parametrized, and tree distances do not depend on the currently learned feature, we do not expect any problems of using approximated trees for back propagation. We can construct a tree based on learned features: for example, we can apply QuadTree to the learned feature to make tree weights depend on the parameters. However, this tree will not be as informative as injecting external structural knowledge to constrain the representation geometry, which is the major focus of structured representation learning.
> * Besides, there are many ways to construct trees. Even when the hierarchy does not exist in the original dataset, we can extract information from other knowledge bases such as WordNet [1], or using LLMs to generate label hierarchies [2].
> * We can also use “multiple” approximated trees in parallel for OT-CPCC learning, which is reasonable if we want to have a more accurate approximation of the ideal tree. There are two scenarios. First, if all approximated trees have the same structure, we can merge all trees into one by using a new tree with the same structure, but using the edge weights averaged from all trees. Then this problem reduces to the standard OT-CPCC problem. Second, if approximated trees have different structures, we can leverage algorithms used in Federated Learning. For example, if we use FederatedSGD, for each tree, we can get a copy of gradient w.r.t. the model parameters. Because these gradients have the same dimension, the final aggregated gradient update can be set as the average of the gradient for each tree.
> In summary, OT-CPCC methods do not rely on the existence of an ideal tree. Any tree can be used in the learning pipeline optimized by backpropagation without any issue.
>
> [1] Miller, George A. "WordNet: a lexical database for English." Communications of the ACM 38.11 (1995): 39-41.
>
> [2] Wan, Mengting, et al. "Tnt-llm: Text mining at scale with large language models." Proceedings of the 30th ACM SIGKDD Conference on Knowledge Discovery and Data Mining. 2024.
>
>
> | Objective |   Setup   |  CPCC  | FineAcc | CoarseAcc |
> |-----------|-----------|--------|---------|-----------|
> |           | depth = 3 |  88.80 |  71.91  |   82.21   |
> |    **EMD**    | depth = 4 |  85.09 |  72.59  |   82.93   |
> |           | depth = 5 |  88.64 |  72.20  |   82.50   |
> |           | depth = 3 |  90.75 |  72.71  |   83.31   |
> |   **FastFT**  | depth = 4 |  87.86 |  72.45  |   83.05   |
> |           | depth = 5 |  90.27 |  72.56  |   83.04   |

---

> ### Author Response · Authors · 2024-11-22
> **Reply to Reviewer 1XYv: Part 3**
>
> > *Finally, a literature review on learning label representations, accompanied by illustrative examples, would provide valuable context and motivation for the study. This would help readers appreciate the importance of this approach within the broader scope of label representation learning.*
>
> Thanks for mentioning label representation learning! However, we believe that OT-CPCC methods are not label representation learning methods. In label representation learning, each label is mapped to some high dimensional vector for downstream tasks: for example in the standard multi-class classification, we typically use one-hot vector embeddings. Alternative choices include using a soft label to assign some probability values in (0,1) on each class [1]. While $\ell_2$-CPCC falls into this category, OT-CPCC methods are not label embedding methods. Instead of representing one class with an Euclidean centroid in $\ell_2$, OT-CPCC methods intentionally avoided this step by considering a more nuanced relationship between labels. OT-CPCC depends on pairwise relationships between data points from two classes, to include a more complicated distributional geometry during learning. Since we rely on data point relations, there is no one-to-one correspondence between a label and a representation.
>
> We include the difference between label representation/embedding learning and structured representation learning in App.B of the current version.
>
> [1] Nguyen, Quang, Hamed Valizadegan, and Milos Hauskrecht. "Learning classification models with soft-label information." Journal of the American Medical Informatics Association 21.3 (2014): 501-508.

---

### Author Response · Authors · 2024-11-22
**Summary of Rebuttal**

We sincerely thank all the reviewers for providing constructive feedback on our paper. In the current submission, we highlighted all the new revisions in blue. In summary, our rebuttal mainly includes
* Several clarifications and revisions about Proposition 3.1, Theorem 3.2, and Lemma 3.4, and time complexity of CPCC algorithms in Tab.1 to address the concern from Reviewer *1XYv*, *rpnu*, and *Tr4V*. The revision for each section is located at:
	* line 181-183 for rephrasing of Prop 3.1’s statement
	* line 210-214 and 854-861 for clarification and detailed proof sketch of Theorem 3.2
	* line 295-297 for applicability of Lemma 3.4 on Fast FlowTree
	* App.I for explanation of time complexities in Tab.1
* Comparison with other related fields including learning label representations/embeddings and extreme classifications as mentioned by Reviewer *1XYv* and *LYtZ*. We include this discussion in App.B.
* Additional experiments including:
    * Effect of Tree Depth on OT-CPCC methods in App.H.1(with additional metrics) and Effect of Tree Weights in App.H.2. These two sets of experiments show that OT-CPCC methods can embed all different kinds of tree information well, and can be generalized to approximated trees in practice, a scenario mentioned by Reviewer *1XYv*.
    * Wall clock run time comparison for OT-CPCC methods training for one epoch to answer the questions from Reviewer *rpnu* and *BGq8*. All numbers are averaged over three random seeds. The results are shown in the very end of this comment, and we put it in App.K of the current version. We can observe that in different empirical datasets, the relationship between different OT-CPCC methods matches our theoretical analysis in Tab.1 and Fig.3 on synthetic data. Along with all other time complexity analysis, the wall clock time comparison further supports the scalability of our FastFT-CPCC.

| Dataset  | Objective | Per Epoch (s)  | Dataset  | Objective | Per Epoch (s)  |
|----------|-----------|----------------|----------|-----------|----------------|
|          | Flat      | 85.88 (3.69)   |          | Flat      | 147.67 (4.54)  |
|          | $\ell_2$        | 83.78 (9.80)   |          | $\ell_2$        | 149.39 (1.24)  |
| **CIFAR10**  | FastFT    | 85.21 (4.26)   | **INAT**     | FastFT    | 132.01 (3.24)  |
| *(b=128)*  | EMD       | 87.70 (2.43)   | *(b=1024)* | EMD       | 234.68 (4.40)  |
|          | Sinkhorn  | 85.65 (4.15)   |          | Sinkhorn  | 165.69 (4.60)  |
|          | SWD       | 92.90 (4.65)   |          | SWD       | 152.52 (8.90)  |

---

### Meta-Review · Area_Chair_6Ts3 · 2024-12-09

**Metareview:**

$\ell_2$-CPCC has been used to embed structured knowledge within labels into feature representations, but has the limitation that it may misrepresent class relationships, especially in multi-modal cases. To address this limitation, the paper propose a EMD-CPCC that use earth mover's distance instead of using class means. Since EMD has high computational cost, the paper further proposes Fast FlowTree which is a linear-time approximation method. Experiments show the benefits of the proposed methods.

The reviewers point out that the paper is well-written and easy to follow, the proposed method is novel and scalable, the experiments are extensive and demonstrates the improvement, and the paper has strong theoretical analysis.

The main concerns of the reviewers were:
- Theory/technical details: proofs are vague, can be simplified, or can be clarified further
- Novelty is lacking: this is a follow-up to the CPCC work by Zeng et al. (2022)
- EMD-based computation is expensive when we have many samples in each class or when we have many classes
- Experiments: FastFT sometimes fails to outperform the baseline or gains are modest
- Suggestions for more contents: literature review on label representation learning, experiments on extreme classification datasets, results on computation time

The authors have updated the paper to address most of the concerns directly, or they have added discussions about them in the paper.

Overall, the reviewers feel positive and the average rating of this paper became higher after the discussions. I recommend accepting this paper to the conference.

**Additional Comments On Reviewer Discussion:**

After the rebuttal and discussions, two reviewers felt their concerns were addressed and have raised their scores. Finally, 4 reviewers have a positive score (6,6,8,8) and 1 reviewer has a negative borderline score (5). The reviewer with 5 did not provide a response to the rebuttal of the authors (at least at the time of writing the meta-review).

---

### Decision · Program_Chairs · 2025-01-22

Accept (Poster)